# Combined forces of hydrostatic pressure and actin polymerization drive endothelial tip cell migration and sprouting angiogenesis

Igor Kondrychyn[1], Liqun He[2], Haymar Wint[1], Christer Betsholtz[2,3], Li-Kun Phng[1]*

[1]Laboratory for Vascular Morphogenesis, RIKEN Center for Biosystems Dynamics Research, Kobe, Japan; [2]Department of Immunology, Genetics and Pathology, Rudbeck Laboratory, Uppsala University, Uppsala, Sweden; [3]Department of Medicine Huddinge, Karolinska Institutet, Huddinge, Sweden

## eLife Assessment

This study **convincingly** shows that aquaporin-mediated cell migration plays a key role in blood vessel formation during zebrafish development. In particular, the article implicates hydrostatic pressure and water flow as mechanisms controlling endothelial cell migration during angiogenic sprouting. This **fundamental** study is highly novel and significantly advances our understanding of cell migration during morphogenesis. As such, this work will be of great interest to developmental and cell biologists working on organogenesis, angiogenesis, and cell migration.

*For correspondence:
likun.phng@riken.jp

Competing interest: The authors declare that no competing interests exist.

**Abstract** Cell migration is a key process in the shaping and formation of tissues. During sprouting angiogenesis, endothelial tip cells invade avascular tissues by generating actomyosin-dependent forces that drive cell migration and vascular expansion. Surprisingly, endothelial cells (ECs) can still invade if actin polymerization is inhibited. In this study, we show that endothelial tip cells employ an alternative mechanism of cell migration that is dependent on Aquaporin (Aqp)-mediated water inflow and increase in hydrostatic pressure. In the zebrafish, ECs express *aqp1a.1* and *aqp8a.1* in newly formed vascular sprouts in a VEGFR2-dependent manner. Aqp1a.1 and Aqp8a.1 loss-of-function studies show an impairment in intersegmental vessels formation because of a decreased capacity of tip cells to increase their cytoplasmic volume and generate membrane protrusions, leading to delayed tip cell emergence from the dorsal aorta and slower migration. Further inhibition of actin polymerization resulted in a greater decrease in sprouting angiogenesis, indicating that ECs employ two mechanisms for robust cell migration in vivo. Our study thus highlights an important role of hydrostatic pressure in tissue morphogenesis.

## Introduction

The blood vascular system is a dynamic, multicellular tissue that adapts to the metabolic demands of development, growth, and homeostasis by altering its pattern and morphology. Macroscopic transformation in the vascular network is achieved by microscopic changes in endothelial cell (EC) behaviors. For example, vascular expansion through sprouting angiogenesis requires the coordination of EC migration, proliferation, anastomosis, lumen formation, and cell rearrangements, all of which require specific cell shape changes. At the initial phase of sprouting angiogenesis, endothelial tip cells generate numerous membrane protrusions such as filopodia and lamellipodia that drive migration and

anastomosis (*Figueiredo et al., 2021*; *Gerhardt et al., 2003*; *Phng et al., 2013*; *Sauteur et al., 2017*). During tubulogenesis, apical membranes deform by generating inverse blebs to expand lumens so that blood can be transported efficiently through the blood vascular network (*Gebala et al., 2016*). However, following vessel perfusion, ECs become less dynamic and instead develop mechanoresistance to the deforming forces of blood pressure to maintain vessel morphology (*Kondrychyn et al., 2020*). EC shape is therefore dynamic and changes depending on the morphogenetic state of vascular development.

It is well established that localized remodeling of actin cytoskeleton and non-myosin II contractility are instrumental in driving cell shape changes during tissue morphogenesis (*Clarke and Martin, 2021*; *Munjal and Lecuit, 2014*; *Murrell et al., 2015*). More recently, there is an accumulating body of work implicating a role of water flow and the resulting increase in hydrostatic pressure as another mechanical mechanism of cell shape change (*Choudhury et al., 2022*; *Chugh et al., 2022*; *Li et al., 2020*). For example, in the osmotic engine model of cell migration, polarized distribution of ion channels and transporters generates an osmotic gradient that drives water inflow at the leading edge and outflow at the rear, leading to forward translocation of tumor cells in confined microenvironment (*Stroka et al., 2014*). Additionally, hydrostatic pressure from the extracellular environment can drive tissue morphogenesis, as demonstrated during mouse blastocyst formation (*Chan et al., 2019*; *Dumortier et al., 2019*), development of the otic vesicle in the zebrafish (*Mosaliganti et al., 2019*), and blood vessel lumen expansion (*Gebala et al., 2016*). Hydrostatic pressure can therefore generate sufficient forces to shape tissues and organs to their proper form and size during development.

We have previously discovered that EC migration persists after the inhibition of actin polymerization and in the absence of filopodia to generate new blood vessels in the zebrafish (*Phng et al., 2013*). This observation suggests the existence of an alternative mechanism of migration independent of actin polymerization. In this study, we sought to determine whether hydrostatic pressure regulates EC migration during sprouting angiogenesis in the zebrafish by investigating the function of Aquaporins (Aqp), which are transmembrane water channels that increase water permeability of cell membranes to promote transcellular water flow (*Day et al., 2014*; *Farinas et al., 1997*; *Kozono et al., 2002*; *Preston et al., 1992*). We discovered that ECs of newly formed intersegmental vessels (ISVs) express *aqp1a.1* and *aqp8a.1* mRNA, and observed the enrichment of Aqp1a.1 and Aqp8a.1 proteins in the leading edge of migrating tip cells. Detailed single-cell analyses showed that endothelial tip cells lacking *aqp1a.1* and *aqp8a.1* expression have reduced cell volume, membrane protrusions, and migration capacity. As a result, there is defective sprouting angiogenesis and ISV formation in *aqp1a.1;aqp8a.1* double mutant zebrafish. Notably, when actin polymerization is inhibited in *aqp1a.1;aqp8a.1* double mutant zebrafish, there is a greater impairment in EC migration and sprouting angiogenesis, demonstrating the additive function of actin polymerization and hydrostatic pressure (that is created by water inflow) in generating membrane protrusions to drive EC migration in vivo.

## Results

### *aqp1a.1* and *aqp8a.1* are differentially expressed by tip and stalk cells during sprouting angiogenesis

Of the 18 *aquaporin* genes that are expressed in zebrafish (*Tingaud-Sequeira et al., 2010*), *aqp1a.1* and *aqp8a.1* mRNA have been detected in blood ECs (*Koun et al., 2016*; *Rehn et al., 2011*). We confirmed the endothelial expression of *aqp1a.1* and *aqp8a.1* at 30 hr post fertilization (hpf) and 2, 3, and 4 days post fertilization (dpf) by whole-mount in situ hybridization (*Figure 1—figure supplement 1*). The two endothelial *aqp* genes, however, differ in spatial expression. While *aqp1a.1* is widely expressed in blood vessels of the head and trunk (dorsal aorta [DA], caudal artery [CA], ISVs, dorsal longitudinal anastomotic vessels [DLAV] and caudal vein plexus) at 30 hpf, *aqp8a.1* expression is absent in cerebral blood vessels and is restricted to the DA, CA, and the ventral regions of ISVs in the trunk. After 1 dpf, *aqp8a.1* expression expands to the entire ISV and DLAV. Additionally, the expression of *aqp1a.1* and *aqp8a.1* gradually decreases in the DA and CA so that both are absent by 4 dpf. Single-cell RNA sequencing (scRNAseq) of ECs isolated at 24 hpf, 34 hpf, and 3 dpf further confirmed the endothelial expression of *aqp1a.1* and *aqp8a.1* mRNA (*Figure 1—figure supplement 2A–F*), revealed differential expression of *aqp1a.1* and *aqp8a.1* mRNA in distinct endothelial clusters, and

highlighted elevated *aqp1a.1* transcript expression in all endothelial subtypes compared to *aqp8a.1* (*Figure 1—figure supplement 2E–H*).

We next examined the expression pattern of *aqp1a.1* and *aqp8a.1* at the beginning of sprouting angiogenesis at higher resolution using RNAscope and confocal microscopy. At 20 hpf, *aqp1a.1* mRNA is highly expressed in the DA with little expression in the posterior cardinal vein (PCV) (*Figure 1A and C*, *Figure 1—figure supplement 3*). Closer inspection uncovered heterogeneous expression in the DA, with higher expression in ECs that would be specified as tip cells (*Figure 1*, *Figure 1—figure supplement 3*). Indeed, tip cells of newly formed vascular sprouts express higher levels of *aqp1a.1* compared to adjacent ECs in the DA at 20 hpf (*Figure 1A–C*, *Figure 1—figure supplement 3B*) and the majority of ISVs (61%) display higher *aqp1a.1* expression in tip cells than in trailing stalk cells at 22 hpf (*Figure 1G*). *Aqp8a.1* expression is at first largely absent in the DA at 20 hpf (*Figure 1B and D*, *Figure 1—figure supplement 3C*) but becomes detectable at 22 hpf (*Figure 1E and F*). Interestingly, unlike *aqp1a.1*, *aqp8a.1* expression is undetectable in tip cells of emerging ISVs at 20 hpf. At 22 hpf, it is expressed in 22 out of 23 tip cells analyzed, with 78% of ISVs exhibiting higher *aqp8a.1* mRNA expression in stalk cells than tip cells (*Figure 1G*). These observations highlight differential expression pattern of *aqp1a.1* and *aqp8a.1* in newly formed vascular sprouts, with *aqp1a.1* expression enriched in tip cells and *aqp8a.1* expression in stalk cells.

## VEGF-VEGFR2 signaling induces the expression of *aqp1a.1* and *aqp8a.1* mRNA

We next explored whether *aqp1a.1* and *aqp8a.1* expression is regulated by VEGFR2 signaling, which is highly activated in tip cells during sprouting angiogenesis (*Gerhardt et al., 2003*; *Siekmann et al., 2013*). To inhibit VEGFR2 activity, embryos were treated with 1 μM ki8751 from 20 hpf. Relative quantitative PCR (qPCR) experiments showed that *aqp1a.1* and *aqp8a.1* mRNA expression was suppressed by 52% (p<0.0001) and 17% (p=0.0028), respectively, after 6 hr of ki8751 treatment compared to DMSO-treated embryos (*Figure 1H*). These results demonstrate that VEGFR2 activity upregulates the expression of *aqp1a.1* and *aqp8a.1*. To examine whether VEGFR2 activity also regulates Aquaporin expression in human ECs, we treated cultured human aortic endothelial cells (HAECs) with ki8751 for 6 hr and analyzed *AQP1* expression by qPCR. *AQP8* expression was not examined since it is not expressed in HAECs (data not shown). Inhibition of VEGFR2 activity with 1 nM and 10 nM ki8751 decreased *AQP1* mRNA expression by 31% (p=0.0023, *Figure 1I*) and 40% (Pp0.1032, *Figure 1I*), respectively, demonstrating cross-species conservation of VEGFR2 activity in inducing *Aqp1* mRNA expression.

## Aqp1a.1 and Aqp8a.1 proteins are enriched at the leading edge of migrating tip cells

To gain a better understanding of where within the cell Aquaporin protein functions, we generated stable *Tg(fli1ep:aqp1a.1-mEmerald)*[rk30] and *Tg(fli1ep:aqp8a.1-mEmerald)*[rk31] zebrafish lines in which ECs express Aqp1a.1 or Aqp8a.1 tagged with mEmerald, respectively. At 25 hpf, we observed an accumulation of Aqp1a.1 (*Figure 2A*) and Aqp8a.1 (*Figure 2B*) in mCherryCAAX-positive membrane clusters in the cytoplasm and filopodia of tip cells. Notably, many of the Aqp1a.1- or Aqp8a.1-positive membrane clusters are located at the migrating front of the tip cell. Time-lapse imaging capturing the process of ISV formation between 24 hpf and 30 hpf further showed sustained enrichment of Aqp1a.1 and Aqp8a.1 in clusters at the front of the tip cell as well as in filopodia (*Figure 2C and D*, *Figure 2—videos 1 and 2*), suggesting that Aqp1a.1 and Aqp8a.1 may promote localized water flux at the migrating edge of tip cells.

## Loss of Aqp1a.1 and Aqp8a.1 function leads to defective trunk blood vessel formation

We subsequently proceeded to generate zebrafish *aqp1a.1* and *aqp8a.1* mutants to investigate the function of Aquaporin-mediated water flow in blood vessel development. Using CRISPR/Cas9 genome editing, we targeted the first exons of *aqp1a.1* and *aqp8a.1* genes. The CRISPR-generating allele *aqp1a.1*[rk28] carries a 14 bp deletion and an 8 bp insertion in the 5' end of the gene, leading to a premature termination codon at amino acid 76 after 1 missense amino acid (*Figure 3—figure supplement 1A and B*). The *aqp8a.1*[rk29] allele carries a 4 bp deletion in the 5' end of the gene that leads to

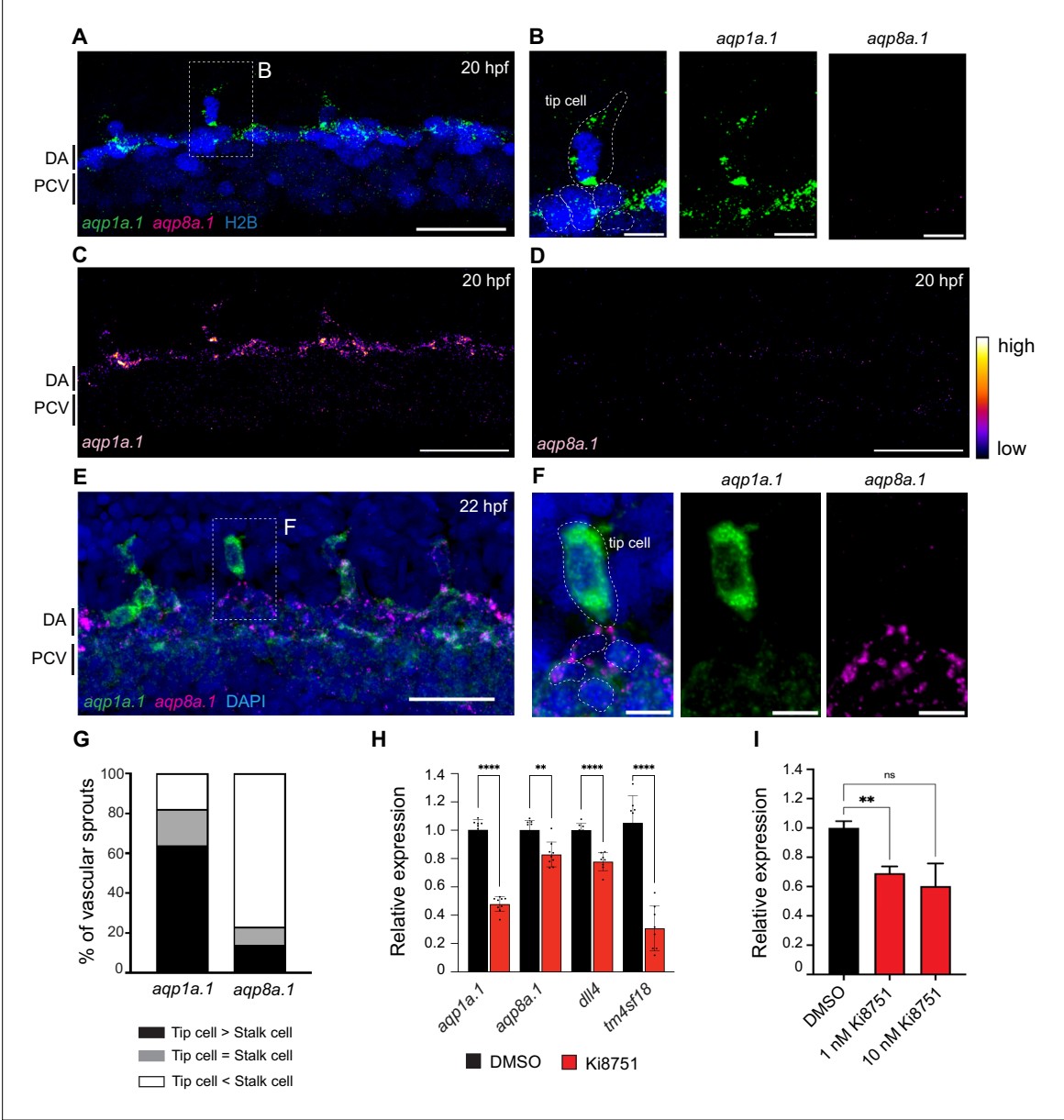

**Figure 1.** Differential expression of *aqp1a.1* and *aqp8a.1* mRNAs in tip and stalk cells. (**A–F**) Detection of *aqp1a.1* and *aqp8a.1* mRNA by RNAscope in situ hybridization in 20 (**A–D**) and 22 (**E, F**) hpf embryos. Representative maximum intensity projection of confocal z-stacks of tip and stalk cells sprouting from DA are shown (n = 9 embryos, two independent experiments). (**B**) and (**F**) show magnified region of a tip cell from a 20 hpf and 22 hpf zebrafish, respectively (tip cell and EC nuclei in the DA are outlined). (**G**) Percentage of vascular sprouts with differential expression of *aqp1a.1* and *aqp8a.1* mRNA in tip and stalk cells in 22 hpf embryos (n = 23 sprouts from eight embryos, two independent experiments). (**H**) qPCR analysis of *aqp1a.1* and *aqp8a.1* expression in zebrafish embryos treated with 0.01% DMSO or 1 µM Ki8751 at 20 hpf for 6 hr (n = 3 independent experiments). (**I**) qPCR analysis of *AQP1* gene expression in human aortic endothelial cells (HAECs) treated with 0.01% DMSO, 1 or 10 nM Ki8751 for 6 hr (n = 2 independent experiments). In panels (**H**) and (**J**), gene expressions are shown relative to *gapdh* expression and data are presented as mean ± SD; statistical significance was determined by Brown–Forsythe and Welch ANOVA tests with Dunnett's multiple-comparisons test; ns p>0.05, **p<0.01, ***p<0.001, and ****p<0.0001. DA, dorsal aorta; PCV, posterior cardinal vein. Scale bar, 10 µm (**B, F**) and 40 µm (**A, C–E**).

The online version of this article includes the following source data and figure supplement(s) for figure 1:

**Source data 1.** Raw data used to generate panels G-I.

**Figure supplement 1.** Expression of endothelial-specific *aquaporins* during development.

**Figure supplement 2.** scRNA-seq analysis of *kdrl*-positive cells from 24 hpf, 34 hpf, and 3 dpf zebrafish.

**Figure supplement 3.** *aqp1a.1* and *aqp8a.1* are differentially expressed in the dorsal aorta at 20 hpf.

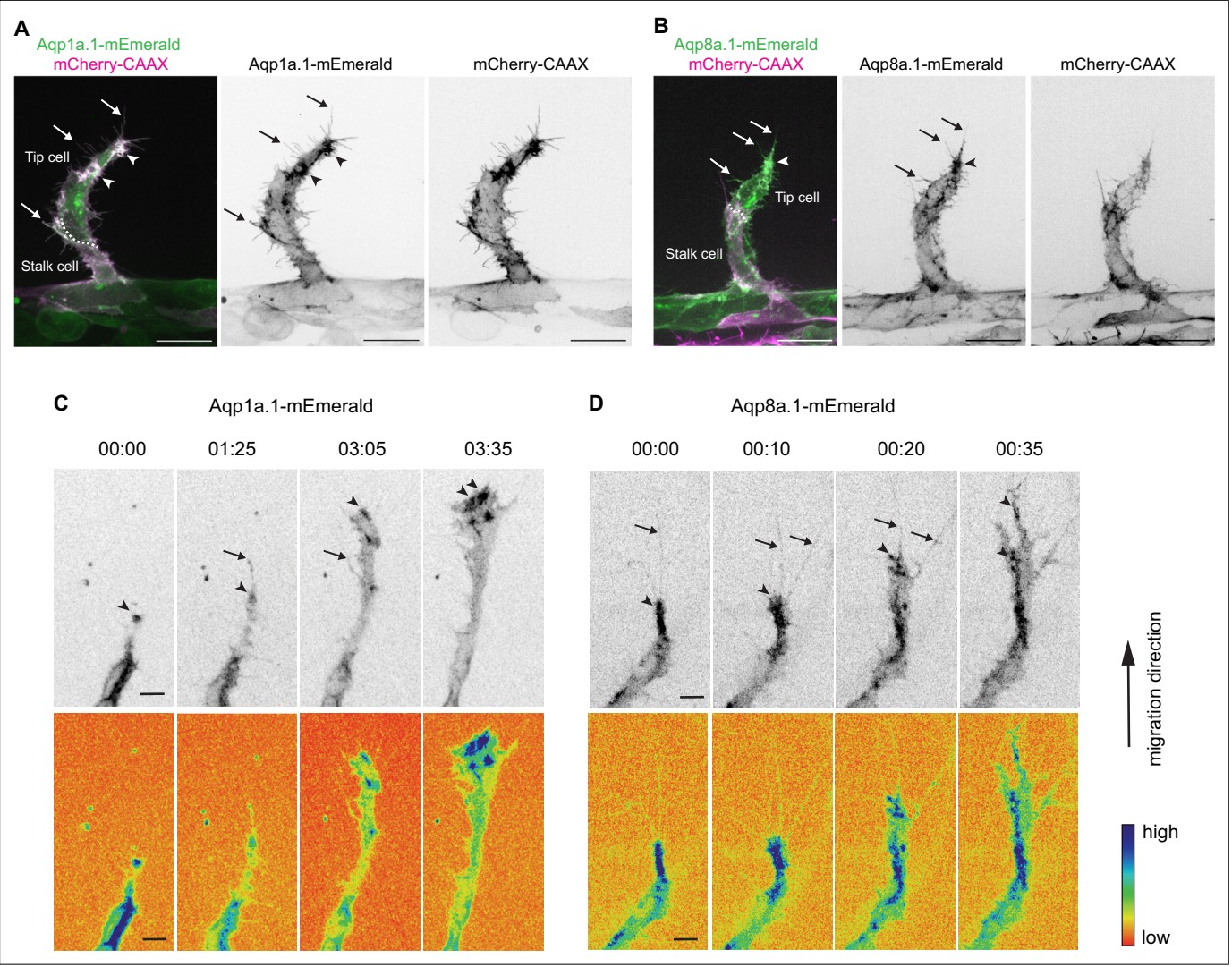

**Figure 2.** Aquaporin proteins are enriched at the leading edge of migrating tip cells. (**A, B**) Representative maximum intensity projection confocal z-stacks of endothelial tip cells of 25 hpf *Tg(fli1ep:aqp1a.1-mEmerald)^{rk30};(kdrl:Hsa.HRAS-mCherry)^{s916}* (**A**) and *Tg(fli1ep:aqp8a.1-mEmerald)^{rk31};(kdrl:Hsa. HRAS-mCherry)^{s916}* (**B**) embryos. White serrated line outlines tip-stalk cell border. (**C, D**) Still images from time-lapse movies of migrating tip cells from *Tg(fli1ep:aqp1a.1-mEmerald)^{rk30}* (**C**) and *Tg(fli1ep:aqp8a.1-mEmerald)^{rk31}* (**D**) embryos. Movies were taken from 24 hpf to 30 hpf. Arrowheads, aquaporin protein localization at the leading edge of migrating tip cells. Arrows, aquaporin protein localization in filopodia. Time, hour:minutes. Scale bar, 5 μm (**C, D**) and 20 μm (**A, B**).

The online version of this article includes the following video(s) for figure 2:

**Figure 2—video 1.** Aqp1a.1 protein enrichment at the leading edge of migrating tip cells.
https://elifesciences.org/articles/98612/figures#fig2video1

**Figure 2—video 2.** Aqp8a.1 protein enrichment at the leading edge of migrating tip cells.
https://elifesciences.org/articles/98612/figures#fig2video2

a frameshift after Thr15 and premature termination codon at amino acid 73 after 58 missense amino acids (*Figure 3—figure supplement 2A and B*). Using whole-mount in situ hybridization and qPCR analysis, we demonstrated that *aqp1a.1* and *aqp8a.1* mRNA expression is significantly decreased in *aqp1a.1^{rk28/rk28}* (*Figure 3—figure supplement 1C and D*) and *aqp8a.1^{rk29/rk29}* (*Figure 3—figure supplement 2C and D*) zebrafish embryos, respectively, suggesting a rapid degradation of mutant mRNAs and as a result, the loss of Aquaporin protein expression.

Although *aqp1a.1^{rk28/rk28};aqp8a.1^{rk29/rk29}* embryos look morphologically normal compared to wildtype or *aqp1a.1^{+/rk28};aqp8a.1^{+/rk29}* embryos (*Figure 3—figure supplement 5A–D*), we observed defects in blood vessel formation and patterning at 1–3 dpf. Compared to 28 hpf control embryos whose ISVs

have reached the dorsal roof of the neural tube and started to form the DLAV, ISVs of *aqp1a.1rk28/rk28;aqp8a.1rk29/rk29* embryos are much less developed (*Figure 3A and D*) with significantly reduced length (*Figure 3E*, *Figure 3—figure supplement 3H and K*). Single deletion of *aqp1a.1* (*Figure 3B*, *Figure 3—figure supplement 3F and I*) or *aqp8a.1* (*Figure 3C*, *Figure 3—figure supplement 3G and J*) also led to defective ISV formation, although the defects are less severe compared to the loss of both *aqp1a.1* and *aqp8a.1* expression. Comparison of ISV length in *aqp1a.1rk28/rk28*, *aqp8a.1rk29/rk29*, and *aqp1a.1rk28/rk28;aqp8a.1rk29/rk29* embryos shows that the loss of *aqp1a.1* leads to a greater decrease in ISV length than the loss of *aqp8a.1* (*Figure 3E*, *Figure 3—figure supplement 3I and J*), and that the loss of both *aqp1a.1* and *aqp8a.1* results in the greatest reduction. These observations indicate an additive effect of Aqp1a.1 and Aqp8a.1 function in blood vessel development and implicate Aqp1a.1 as the more dominant Aquaporin protein in EC function, in line with the higher expression of *aqp1a.1* mRNA in tip cells.

We next examined the embryos at 3 dpf, a stage when arterial ISVs (aISVs) and venous (vISVs) are fully established and lumenized, connecting the DLAV to either the DA or PCV (*Figure 3F and G*), respectively. Such fully established ISVs are significantly reduced in embryos when *aqp1a.1*, *aqp8a.1* or both *aqp1a.1* and *aqp8a.1* are deleted. Instead, these embryos display an increased number of aISVs and vISVs that fail to establish a connection to the DLAV, DA, or PCV (*Figure 3H–J*). The number of incompletely formed or truncated ISVs per embryo in *aquaporin* mutants was variable, between 1–6, 1–16, and 1–22 truncated ISVs in *aqp8a.1rk29/rk29*, *aqp1a.1rk28/rk28* and *aqp1a.1rk28/rk28;aqp8a.1rk29/rk29* embryos, respectively (*Figure 3—figure supplement 3A–D*). We also found that truncated ISVs appear with higher frequency in the zebrafish trunk (ISV number 5–15) rather than in the tail (ISV number 16–27, *Figure 3—figure supplement 3E*). Detailed analysis revealed that most of the truncated ISVs are vISVs that are connected to the PCV but not the DLAV (Class I in *Figure 3K*, *Figure 3—figure supplement 4A*), followed by the absence of aISV or vISV (Class II in *Figure 3K*, *Figure 3—figure supplement 4B*). A small fraction of the truncated vessels are aISVs that are not connected to the DA (Class IV in *Figure 3K*, *Figure 3—figure supplement 4D*) or DLAV (Class V in *Figure 3K*, *Figure 3—figure supplement 4E*), or vISVs that are not connected to the DLAV (Class VI in *Figure 3K*, *Figure 3—figure supplement 4F*). At some regions, there is a truncated aISV that is not connected to the DA and an incomplete vISV that is not connected to the DLAV (Class III in *Figure 3K*, *Figure 3—figure supplement 4C*).

As *aqp1a.1* is expressed in the brain vasculature during development (*Figure 1—figure supplement 1*), we examined whether cerebral vasculature is also affected in *aqp1a.1rk28/rk28* embryos. At 4 dpf, we found that the length of the dorsal longitudinal vein (DLV), a main blood vessel that supplies blood to the choroid plexus (*Bill and Korzh, 2014*), was reduced in *aqp1a.1rk28/rk28* embryos (p=0.0098, *Figure 3—figure supplement 6A, B and E*). Also, in 2 out of 21 *aqp1a.1rk28/rk28* embryos, the DLV was undeveloped (*Figure 3—figure supplement 6C, D and F*), a defect that is also observed in 50% of *vegfab-/-* and *vegfc-/-* mutant zebrafish (*Parab et al., 2021*).

In summary, our analyses suggest that in the absence of Aqp1a.1 and/or Aqp8a.1 function, there is an initial delay in ISV formation during primary sprouting angiogenesis. By 3 dpf, some ISVs are completely formed (*Figure 3—figure supplement 5K and M*), connecting the DA or PCV to the DLAV. However, the number of completely formed ISVs is significantly decreased, with the loss of Aqp1a.1 having a more severe impact compared to the loss of Aqp8a.1, and the combined loss of Aqp1a.1 or Aqp8a.1 having a greater deleterious effect.

## Aquaporins promote endothelial tip cell protrusion and migration

To gain a better understanding of why ISVs do not develop normally, we assessed EC number. By counting the number of nuclei in aISVs and vISVs, we found a significant reduction in EC number per vessel in 2 dpf and 3 dpf *aqp1a.1rk28/rk28;aqp8a.1rk29/rk29* embryos compared to wildtype embryos (*Figure 4—figure supplement 1A and B*), suggesting that the defective formation of ISVs may result from insufficient EC number. Quantification of endothelial mitotic events between 21 hpf and 30 hpf shows that there is a similar number of tip and stalk cell division in *aqp1a.1+/rk28;aqp8a.1+/rk29* embryos and *aqp1a.1rk28/rk28;aqp8a.1rk29/rk29* embryos (*Figure 4—figure supplement 1C*), indicating that decreased EC division is not the cause of reduced EC number in ISVs. We next explored whether EC migration from the DA is impaired by examining tip cell behavior from 21 hpf to 22 hpf, when sprouting angiogenesis begins. In *aqp1a.1+/rk28;aqp8a.1+/rk29* embryos, tip cells emerge from the DA

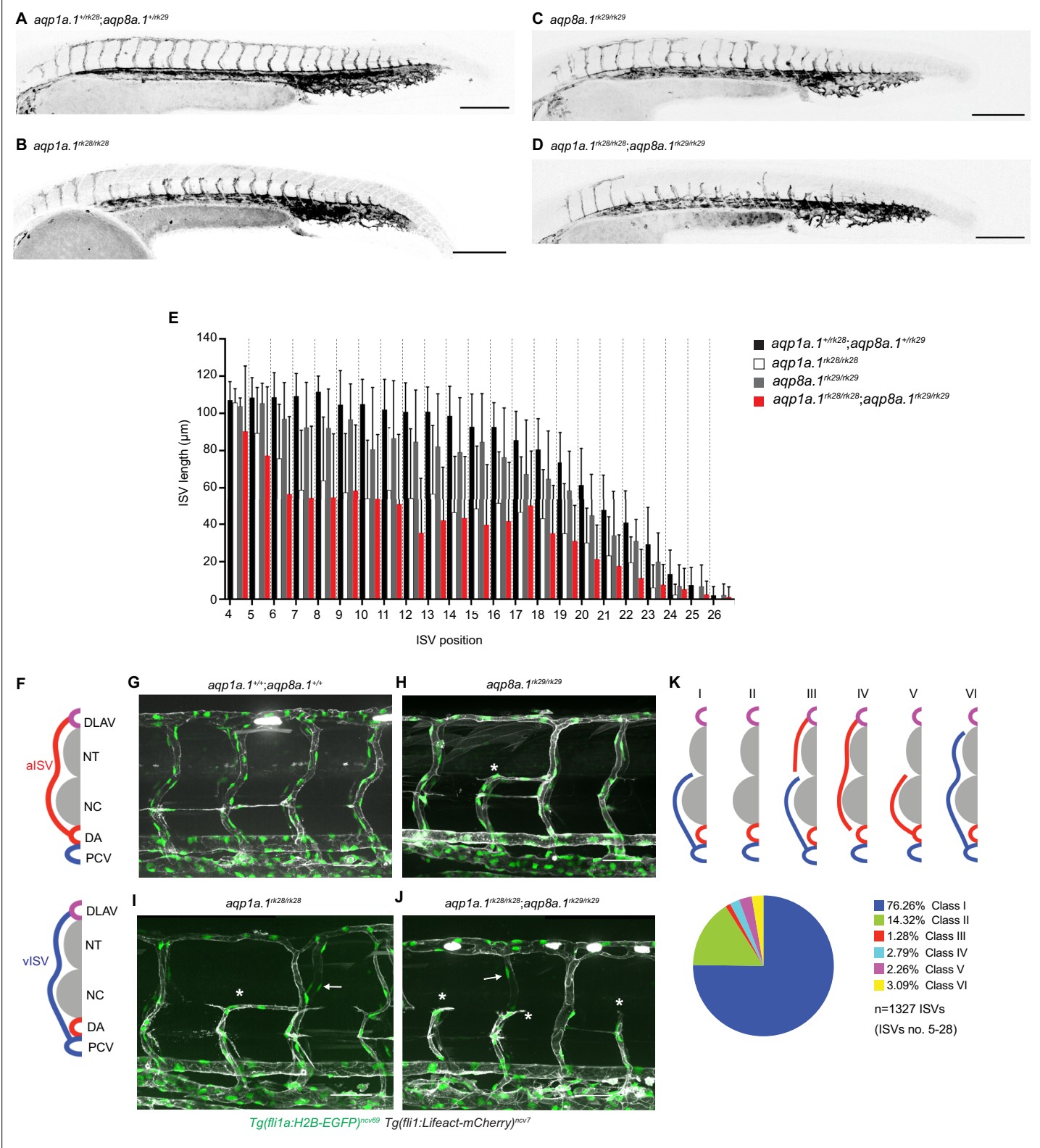

**Figure 3.** Loss of Aquaporin function leads to defective trunk vessel formation. (**A–D**) Representative maximum intensity projection confocal z-stacks of 28 hpf *aqp1a.1*[+/rk28]*;aqp8a.1*[+/rk29] (**A**), *aqp1a.1*[rk28/rk28] (**B**), *aqp8a.1*[rk29/rk29] (**C**), and *aqp1a.1*[rk28/rk28]*;aqp8a.1*[rk29/rk29] (**D**) embryos. (**E**) Quantification of ISV length at different positions along the trunk in 28 hpf embryos (*aqp1a.1*[+/rk28]*;aqp8a.1*[+/rk29]: n = 21 embryos; *aqp1a.1*[rk28/rk28]: n = 20 embryos; *aqp8a.1*[rk29/rk29]: n = 20 embryos; *aqp1a.1*[rk28/rk28]*;aqp8a.1*[rk29/rk29]: n = 19 embryos from two independent experiments); data is presented as mean ± SD. (**F**) Illustration of aISV

*Figure 3 continued on next page*

*Figure 3 continued*

and vISV connections in the trunk of wildtype embryo at 3 dpf. (**G–J**) Representative maximum intensity projection confocal z-stacks of 3 dpf wild type (**G**), *aqp8a.1$^{rk29/rk29}$* (**H**), *aqp1a.1$^{rk28/rk28}$* (**I**), and *aqp1a.1$^{rk28/rk28}$;aqp8a.1$^{rk29/rk29}$* (**J**) embryos. Asterisk marks incomplete ISVs, arrows point to EC nuclei on the contralateral side. (**K**) Pie chart showing the proportion of different classes of truncated ISV phenotypes (I–VI) found in *aqp1a.1$^{rk28/rk28}$;aqp8a.1$^{rk29/rk29}$* mutant embryos at 3 dpf. DA, dorsal aorta; DLAV, dorsal longitudinal anastomotic vessel; ISV, intersegmental vessel; aISV, arterial ISV; vISV, venous ISV; NC, notochord; NT, neural tube; PCV, posterior cardinal vein. Scale bar, 50 µm (**G–J**) and 200 µm (**A–D**).

The online version of this article includes the following source data and figure supplement(s) for figure 3:

**Source data 1.** Raw data used to generate panel E.

**Figure supplement 1.** CRISPR/Cas9-induced mutation in zebrafish *aqp1a.1* gene.

**Figure supplement 2.** CRISPR/Cas9-induced mutation in zebrafish *aqp8a.1* gene.

**Figure supplement 3.** Analysis of ISV defects in *aquaporin* mutant embryos.

**Figure supplement 3—source data 1.** Raw data used to generate panels A-E, I-K.

**Figure supplement 4.** ISV phenotype classification in *aqp1a.1$^{rk28/rk28}$;aqp8a.1$^{rk29/rk29}$* mutant embryos at 3 dpf.

**Figure supplement 5.** Phenotype of *aqp1a.1$^{rk28/rk28}$;aqp8a.1$^{rk29/rk29}$* double mutant at 28 hpf and 3 dpf.

**Figure supplement 6.** Cerebral vascular formation defects in *aqp1a.1* mutant zebrafish.

**Figure supplement 6—source data 1.** Raw data used to generate panel E.

**Figure supplement 7.** Depletion of *aqp1a.1* and *aqp8a.1* expression alters vessel diameter.

**Figure supplement 7—source data 1.** Diameter of ISVs in wildtype, aquaporin mutant and aquaporin-overexpressing embryos at 2 dpf.

between 20 hpf and 24 hpf (*Figure 4A*) and migrate dorsally between the somite boundaries over the notochord and neural tube to form primary ISVs (*Figure 4B*, *Figure 4—video 1*). However, in *aqp1a.1$^{rk28/rk28}$;aqp8a.1$^{rk29/rk29}$* embryos, tip cell nuclei emerge between 22 hpf and 28 hpf (*Figure 4A*). Furthermore, in some *aqp1a.1$^{rk28/rk28}$;aqp8a.1$^{rk29/rk29}$* embryos, tip cell membrane protrusions do not extend dorsally in the direction of migration but retract back to the DA so that a primary ISV does not form at this location (*Figure 4C*, *Figure 4—video 2*), potentially giving rise to the Class II phenotype at 3 dpf (*Figure 3K*, *Figure 3—figure supplement 4B*). These observations indicate that tip cell protrusion from the DA is significantly delayed in the absence of Aqp1a.1 and Aqp8a.1 function.

We have previously demonstrated that filopodia facilitate EC migration by serving as a template for lamellipodia formation (*Phng et al., 2013*). Lamellipodia frequently emanate laterally from stable filopodia at the leading edge of tip cells, leading to rapid expansion of the filopodium, increase in volume, and the formation of a stable protrusion in the direction of migration (*Phng et al., 2013*). This was similarly observed in *aqp1a.1$^{+/rk28}$;aqp8a.1$^{+/rk29}$* embryos (*Figure 4D*, *Figure 4—video 3*) but not in *aqp1a.1$^{rk28/rk28}$;aqp8a.1$^{rk29/rk29}$* embryos (*Figure 4E*, *Figure 4—video 4*). Comparison of filopodia dynamics in wildtype (*Figure 4F*), *aqp1a.1$^{+/rk28}$;aqp8a.1$^{+/rk29}$* and *aqp1a.1$^{rk28/rk28}$;aqp8a.1$^{rk29/rk29}$* (*Figure 4G*) embryos revealed a significant reduction in filopodia number (wildtype, 18.73 ± 5.1/100 µm; *aqp1a.1$^{+/rk28}$;aqp8a.1$^{+/rk29}$*, 20.4 ± 6.5/100 µm; *aqp1a.1$^{rk28/rk28}$;aqp8a.1$^{rk29/rk29}$*, 9.8 ± 5.5/100 µm; *Figure 4H*) and length (wildtype, 5.5 ± 2.8 µm; *aqp1a.1$^{+/rk28}$;aqp8a.1$^{+/rk29}$*, 6.0 ± 3.4 µm; *aqp1a.1$^{rk28/rk28}$;aqp8a.1$^{rk29/rk29}$*, 4.5 ± 2.2 µm; *Figure 4I*) in ECs lacking both Aqp1a.1 and Aqp8a.1 function that can be accounted for by a decrease in actin polymerization (*Figure 4J*). By tracking Lifeact-mCherry signal in filopodia, we determined that the average elongation rate of actin as 0.055 ± 0.013 µm per second in wildtype embryos and that this is significantly decreased by 47% to 0.029 ± 0.011 µm per second (p<0.0001) in *aqp1a.1$^{rk28/rk28}$;aqp8a.1$^{rk29/rk29}$* embryos. Analysis of time-lapse images of tip cell behavior further showed slower tip cell membrane expansion at the leading edge (*Figure 4K*), decreased nuclei displacement (*Figure 4L*), and a significant reduction in tip cell migration velocity (*Figure 4M*) in *aqp1a.1$^{rk28/rk28}$;aqp8a.1$^{rk29/rk29}$* embryos. To determine whether the slower EC migration is caused by a loss of front-rear polarity, we examined Golgi position relative to the nucleus in wildtype, *aqp1a.1$^{rk28/rk28}$* and *aqp8a.1$^{rk29/rk29}$* embryos at 24–26 hpf (*Figure 4—figure supplement 1D*). This analysis shows that tip cell polarity is unperturbed in the absence of Aqp1a.1 or Aqp8a.1 function.

In conclusion, our findings demonstrate that Aqp1a.1 and Aqp8a.1 regulate sprouting angiogenesis by promoting endothelial tip cell emergence from the DA, the formation and expansion of membrane protrusions and EC migration.

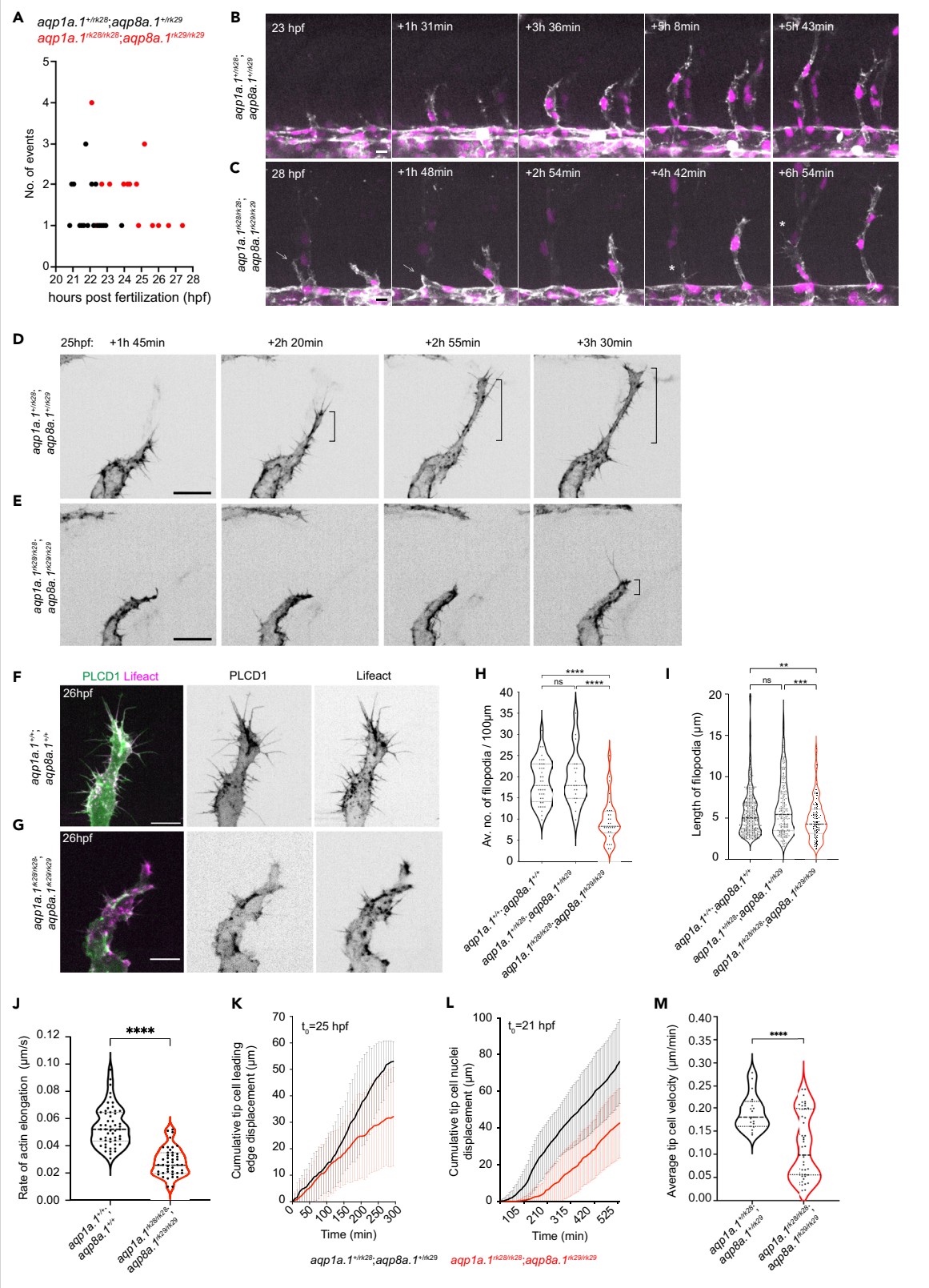

**Figure 4.** Aquaporins promote endothelial tip cell protrusion and migration. (**A**) Timing of tip cell emergence from the dorsal aorta (DA) in *aqp1a.1+/rk28*;*aqp8a.1+/rk29* (n = 23 cells) and *aqp1a.1rk28/rk28*;*aqp8a.1rk29/rk29* (n = 27 cells). (**B, C**) Still images from time-lapse imaging of migrating tip cells from *aqp1a.1+/rk28*;*aqp8a.1+/rk29* (**B**) and *aqp1a.1rk28/rk28*;*aqp8a.1rk29/rk29* (**C**) embryos from 20 hpf to 30 hpf. White arrows, retracting tip cell. *, secondary sprouting from posterior cardinal vein (PCV). Scale bar, 10 µm. (**D, E**) Stills images from representative time-lapse movies of migrating tip cells in *aqp1a.1+/*

*Figure 4 continued on next page*

*Figure 4 continued*

$^{rk28}$;*aqp8a.1*$^{+/rk29}$ (**D**, n = 2) and *aqp1a.1*$^{rk28/rk28}$;*aqp8a.1*$^{rk29/rk29}$ (**E**, n = 5) embryos. Movies were taken from 25 hpf to 30 hpf (n = 2 independent experiments). Bracket, formation of stable protrusions. Scale bar, 20 µm. (**F, G**) Representative maximum intensity projection confocal z-stacks of tip cells from wildtype (**F**) and *aqp1a.1*$^{rk28/rk28}$;*aqp8a.1*$^{rk29/rk29}$ (**G**) embryos at 26 hpf. Scale bar, 10 µm. (**H**) Quantification of filopodia number in tip cells of wildtype (n = 36 cells from 10 embryos, two independent experiments), *aqp1a.1*$^{+/rk28}$;*aqp8a.1*$^{+/rk29}$ (n = 19 cells from six embryos, two independent experiments), and *aqp1a.1*$^{rk28/rk28}$;*aqp8a.1*$^{rk29/rk29}$ (n = 28 cells from nine embryos, two independent experiments) embryos. (**I**) Quantification of filopodia length in tip cells of wildtype (n = 24 cells from seven embryos, two independent experiments), *aqp1a.1*$^{+/rk28}$;*aqp8a.1*$^{+/rk29}$ (n = 16 cells from seven embryos, two independent experiments), and *aqp1a.1*$^{rk28/rk28}$;*aqp8a.1*$^{rk29/rk29}$ (n = 11 cells from six embryos, two independent experiments) embryos. (**J**) Growth rate of actin bundles in tip cell filopodia in 25 hpf wildtype (n = 12 cells from seven embryos, two independent experiments) and *aqp1a.1*$^{rk28}$;*aqp8a.1*$^{rk29}$ (n = 12 cells from six embryos, two independent experiments) embryos. (**K**) Quantification of tip cell leading edge displacement of *aqp1a.1*$^{+/rk28}$;*aqp8a.1*$^{+/rk29}$ (n = 19 cells from five embryos, three independent experiments) and *aqp1a.1*$^{rk28/rk28}$;*aqp8a.1*$^{rk29/rk29}$ (n = 47 cells from 13 embryos, six independent experiments) embryos at 25–30 hpf; data is presented as mean ± SD. (**L**) Quantification of tip cell nuclei displacement of *aqp1a.1*$^{+/rk28}$;*aqp8a.1*$^{+/rk29}$ (n = 20 cells from six embryos, four independent experiments) and *aqp1a.1*$^{rk28/rk28}$;*aqp8a.1*$^{rk29/rk29}$ (n = 20 cells from four embryos, two independent experiments) embryos at 21–30 hpf; data is presented as mean ± SD. (**M**) Quantification of tip cell migration velocity in *aqp1a.1*$^{+/rk28}$;*aqp8a.1*$^{+/rk29}$ (n = 19 cells from five embryos, three independent experiments) and *aqp1a.1*$^{rk28/rk28}$;*aqp8a.1*$^{rk29/rk29}$ (n = 47 cells from 13 embryos, six independent experiments) embryos at 25–30 hpf. Statistical significance was determined by Brown–Forsythe and Welch ANOVA tests with Dunnett's (**H**) or Sidak's (**I**) multiple-comparisons test, and with unpaired *t*-test (**J, M**). ns, p>0.05, **p<0.01, ***p<0.001, and ****p<0.0001.

The online version of this article includes the following video, source data, and figure supplement(s) for figure 4:

**Source data 1.** Raw data used to generate panels A, H-M.

**Figure supplement 1.** Reduction in endothelial cell (EC) number per ISV in *aqp1a.1*$^{rk28/rk28}$;*aqp8a.1*$^{rk29/rk29}$ embryos.

**Figure supplement 1—source data 1.** Raw data used to generate panels A-C.

**Figure 4—video 1.** Sprouting angiogenesis in *aqp1a.1*$^{+/rk28}$;*aqp8a.1*$^{+/rk29}$ heterozygote embryos.
https://elifesciences.org/articles/98612/figures#fig4video1

**Figure 4—video 2.** Defective sprouting angiogenesis in *aqp1a.1*$^{rk28//rk28}$;*aqp8a.1*$^{+rk29/rk29}$ embryos.
https://elifesciences.org/articles/98612/figures#fig4video2

**Figure 4—video 3.** Tip cells form an elongated protrusion in the direction of migration during sprouting angiogenesis.
https://elifesciences.org/articles/98612/figures#fig4video3

**Figure 4—video 4.** Defective tip cell leading edge expansion in *aqp1a.1*$^{rk28/rk28}$;*aqp8a.1*$^{rk29/rk29}$ embryos.
https://elifesciences.org/articles/98612/figures#fig4video4

## Tip cell volume regulation depends on Aquaporin-mediated water influx

Aquaporin channels permit bidirectional flow of water across the plasma membrane, with the direction of flow determined by the osmotic gradient between the extracellular environment and cell cytoplasm (*Agre et al., 2002*). We next sought to determine whether water flows in or out of tip cells during sprouting angiogenesis. We hypothesize that water flux across the cell membrane can lead to changes in tip cell volume, with water influx increasing the volume while outflow decreases. To quantify tip cell volume in the absence of Aquaporin function, we injected a plasmid encoding *kdrl:mEmerald* into one-cell stage *aqp1a.1*$^{rk28/rk28}$;*aqp8a.1*$^{rk29/rk29}$ embryos to label ECs in a mosaic manner. We also investigated the effects of increasing Aqp1a.1 expression on tip cell volume by injecting a plasmid encoding Aqp1a.1-P2A-EGFP into wildtype embryos. As control, wildtype embryos were injected with the plasmid encoding *kdrl:mEmerald*. mEmerald/EGFP-positive tip cells were imaged and their volume measured at 24–25 hpf from 3D reconstructions of labeled cells. Wildtype ECs are on average 964.6 ± 270.2 µm$^3$ in size (*Figure 5A and D*). In the absence of both Aqp1a.1 and Aqp8a.1 function (*Figure 5B*), tip cell volume significantly decreased by 26% to 709.8 ± 293.5 µm$^3$ (p=0.015, *Figure 5D*) compared to wildtype tip cells. The overexpression of Aqp1a.1 increased tip cell volume by 36% to 1313 ± 527.4 µm$^3$ compared to wildtype tip cells (*Figure 5C and D*, p=0.0069) and by 85% compared to *aqp1a.1*$^{rk28/rk28}$;*aqp8a.1*$^{rk29/rk29}$ tip cells (p<0.0001). These results therefore support water influx as the direction of water flow in tip cells during migration, and that this is mediated through Aqp1a.1 and Aqp8a.1.

In summary, our results demonstrate a role of endothelial Aquaporins in promoting water influx to increase endothelial tip cell volume and cytoplasmic hydrostatic pressure, as well as accelerate actin polymerization in filopodia (*Figure 4J*).

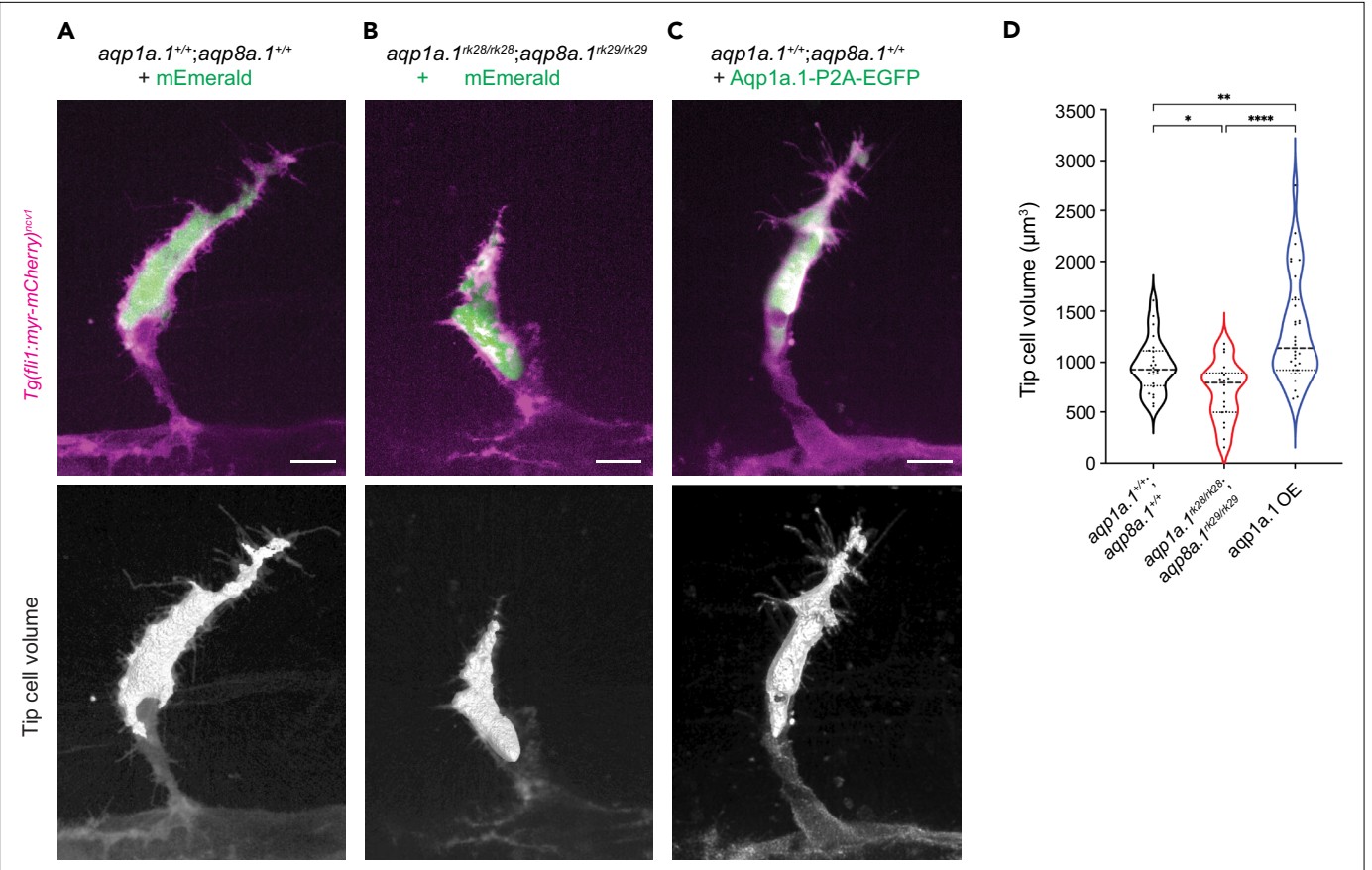

**Figure 5.** Aquaporin-mediated water influx increases tip cell volume. (**A–C**) Representative maximum intensity projection confocal z-stacks of tip cell and its 3D-volume rendering of 25 hpf wildtype (**A**), *aqp1a.1^rk28/rk28;aqp8a.1^rk29/rk29* (**B**) and wildtype embryos overexpressing Aqp1a.1 (**C**) labeled with cytosolic mEmerald or EGFP. Scale bar, 10 μm. (**D**) Quantification of tip cell volume in 25 hpf wildtype (n = 24 cells) and *aqp1a.1^rk28/rk28;aqp8a.1^rk29/rk29* (n = 20 cells) and wildtype overexpressing Aqp1a.1 (n = 32 cells) embryos from three independent experiments. Statistical significance was determined by Brown–Forsythe and Welch ANOVA tests. *p<0.05, **p<0.01, and ****p<0.0001.

The online version of this article includes the following source data for figure 5:

**Source data 1.** Raw data used to generate panel D.

## The anion channel, SWELL1, promotes EC migration and sprouting angiogenesis

As the direction of water flow is dictated by an osmotic gradient, with water flowing from low to high osmolarity, we next examined the function of ion channels in establishing an osmotic gradient. SWELL1, or LRRC8A, is a component of volume-regulated anion channel (VRAC) that mediates ions, especially chloride anions, and osmolyte efflux (*Qiu et al., 2014*; *Voss et al., 2014*). Recently, SWELL1 has been demonstrated to polarize at the trailing edge of migrating breast cancer cells to direct water efflux and confer confined migration direction (*Zhang et al., 2022*). In the zebrafish, scRNAseq data shows that the gene encoding SWELL1, *lrrc8aa*, is expressed in a subset of ECs, with 12.32% of ECs coexpressing *lrrc8aa* and *aqp1a.1* (*Figure 6A*) and 10.16% coexpressing *lrrc8aa* and *aqp8a.1* (*Figure 6B*). To determine whether *lrrc8aa* is expressed in endothelial tip cells, we performed RNAscope and confocal microscopy. Although *lrrc8aa* is expressed in surrounding somites and notochord (*Figure 6C*), we could also detect its expression in endothelial tip cells (*Figure 6D and E*), suggesting that SWELL1 can establish an osmotic gradient in the vicinity of the developing ISVs and within tip cells to direct water flow. To address the function of SWELL1-generated osmotic gradient, we disrupted its activity by treating zebrafish with 5 μM DCPIB (*Gunasekar et al., 2022*) for 6 hr from 20 hpf and observed a similar impairment in ISV formation (*Figure 6F and G*) as in *aqp1a.1^rk28/rk28;aqp8a.1^rk29/rk29* embryos (*Figure 3D and E*). SWELL1 inhibition resulted in shorter ISVs formed at 26–27

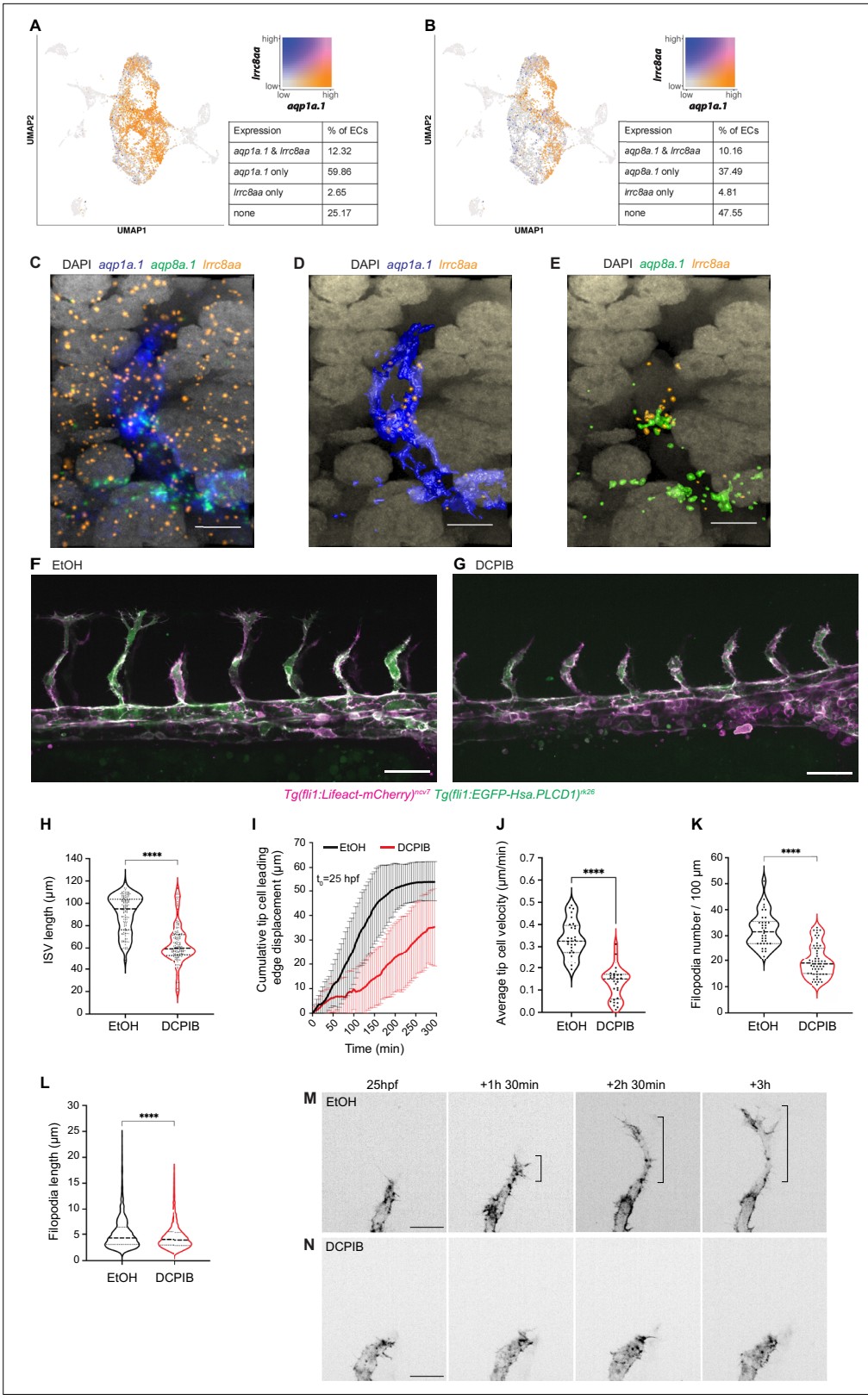

**Figure 6.** The chloride channel SWELL1 promotes endothelial cell (EC) migration and sprouting angiogenesis. (**A, B**) UMAP plots showing co-expression of *lrrc8aa* and *aqp1a.1* (**A**), and *lrrc8aa* and *aqp8a.1* (**B**) in ECs from 24 hpf, 34 hpf, and 3 dpf embryos. (**C–E**) Detection of *aqp1a.1*, *aqp8a.1,* and *lrrc8aa* mRNA by RNAscope in situ hybridization in 22 hpf zebrafish embryo. Representative maximum intensity projection rendering of confocal z-

*Figure 6 continued on next page*

*Figure 6 continued*

stacks of tip and stalk cells (**C**). (**D**) and (**E**) show surface rendering of image (**C**) where only endothelial expression of *lrrc8aa* is shown after masking. Scale bar, 10 μm. (**F, G**) Representative maximum intensity projection confocal z-stacks of 26 hpf wildtype embryos treated with 0.05% EtOH (**F**) or 5 μM DCPIB (**G**) from 20 hpf to 26 hpf (control, n = 16 embryos; DCPIB, n = 18 embryos; two independent experiments). Scale bar, 50 μm. (**H**) Quantification of intersegmental vessel (ISV) length in 26 hpf wildtype embryos treated with 0.05% EtOH or 5 μM DCPIB (control, n = 133 vessels from 16 embryos; DCPIB, 157 vessels from 18 embryos; two independent experiments). (**I**) Quantification of tip cell leading edge displacement of wildtype embryos treated with 0.05% EtOH (n = 29 cells from seven embryos, two independent experiments) or 5 μM DCPIB (n = 29 cells from eight embryos, three independent experiments) at 25–30 hpf; data is presented as mean ± SD. (**J**) Quantification of tip cell migration velocity in wildtype treated with 0.05% EtOH (n = 29 cells from seven embryos, two independent experiments) or 5 μM DCPIB (n = 29 cells from eight embryos, three independent experiments) at 25–30 hpf. (**K**) Filopodia number in tip cells of wildtype embryos treated with 0.05% EtOH (n = 34 vessels from nine embryos, three independent experiments) or 5 μM DCPIB (n = 53 vessels from 16 embryos, three independent experiments). (**L**) Filopodia length in tip cells of wildtype embryos treated with 0.05% EtOH (n = 34 vessels from nine embryos, three independent experiments) or 5 μM DCPIB (n = 53 vessels from 12 embryos, three independent experiments). Statistical significance was determined with unpaired *t*-test; ****p<0.0001 (**H, J, K, and L**). (**M, N**) Still images from representative time-lapse movies of migrating tip cells in wildtype embryos treated with 0.05% EtOH (**M**, n = 7) or 5 μM DCPIB (**N**, n = 8). Movies were taken from 25 to 30 hpf (n = 3 independent experiments). Bracket, formation of stable protrusions. Scale bar, 20 μm.

The online version of this article includes the following video and source data for figure 6:

**Source data 1.** Raw data used to generate panels H-L.

**Figure 6—video 1.** The chloride channel SWELL1 promotes EC migration and sprouting angiogenesis. https://elifesciences.org/articles/98612/figures#fig6video1

hpf (*Figure 6H*) due to impaired leading edge displacement of tip cells (*Figure 6I*) and decreased tip cell velocity (*Figure 6J*). Further analysis of tip cell morphology showed a decrease in the number (*Figure 6K*) and length (*Figure 6L*) of filopodia formed and compromised elongation of tip cells (*Figure 6M and N*, *Figure 6—video 1*) in DCPIB-treated embryos. In summary, these results suggest that chloride efflux through SWELL1 establishes an osmotic gradient, either within ECs or between EC and the surrounding interstitial tissue, to direct water flow and thereby modulate EC migration and sprouting angiogenesis.

## Additive effects of actin polymerization and water influx in driving EC migration and sprouting angiogenesis

We have previously observed that ECs are still able to migrate and form ISVs after the inhibition of actin polymerization (*Phng et al., 2013*). In these experiments, embryos were treated with a low concentration of Latrunculin B (Lat. B), an inhibitor of actin polymerization, that resulted in the suppression of filopodia formation. Under such conditions, ECs can generate small membrane protrusions and migrate in a directed manner at a reduced speed. How ECs are still able to migrate after the inhibition of actin polymerization was unclear. As our current study demonstrates a role of water inflow and hydrostatic pressure as another mechanism of cell migration, we hypothesize that ECs employ water influx to migrate when actin polymerization is compromised and that the depletion of both water influx and actin polymerization will result in a greater inhibition of EC migration.

To test this hypothesis, we examined ISV development in embryos with decreased actin polymerization and water inflow by treating *aqp1a.1*$^{rk28/rk28}$;*aqp8a.1*$^{rk29/rk29}$ embryos with Lat. B. At 28 hpf, ISVs of wildtype embryos treated with DMSO have reached the dorsal roof of the neural tube (*Figure 7A*) and show an average length of 104.1 ± 13.47 μm (*Figure 7E*). The treatment of wildtype embryos with 0.08 μg/ml Lat. B from 20 hpf to 28 hpf (*Figure 7B*) resulted in a significant decrease in ISV length (61.53 ± 23.27 μm) compared to control embryos (p<0.0001, *Figure 7E*). A similar decrease in ISV length was also observed in *aqp1a.1*$^{rk28/rk28}$;*aqp8a.1*$^{rk29/rk29}$ embryos treated with DMSO (63.75 ± 30.24 μm, *Figure 7C and E*). When *aqp1a.1*$^{rk28/rk28}$;*aqp8a.1*$^{rk29/rk29}$ embryos were treated with 0.08 μg/ml Lat. B from 20 hpf to 28 hpf, there was a greater decrease in the length of ISV formed at 28 hpf (43.10 ± 24.42 μm, *Figure 7D and E*) compared to the inhibition of actin polymerization (p=0.0028) or depletion of water influx (p<0.0001) alone (*Figure 7—video 1*). These results therefore demonstrate

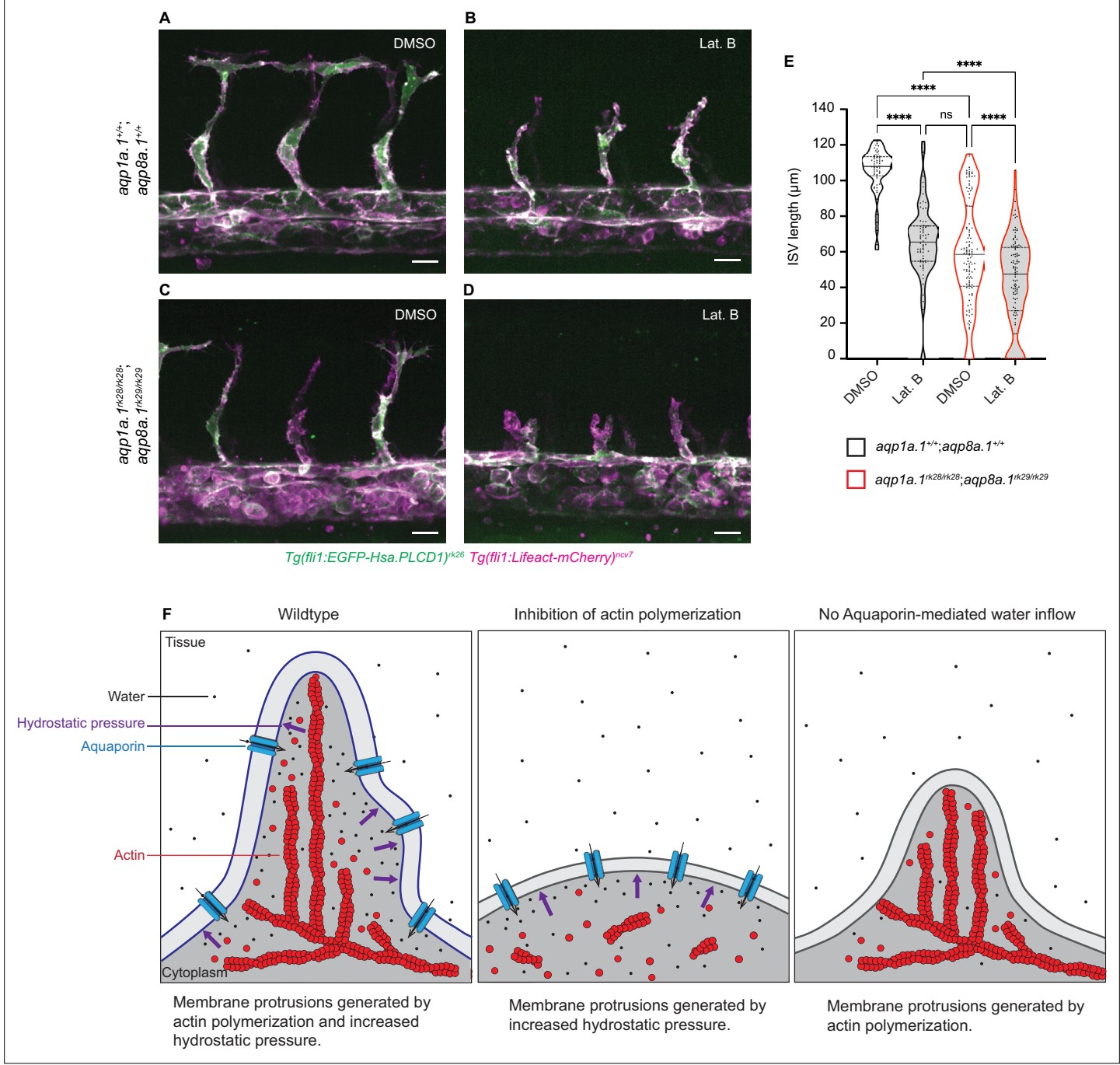

**Figure 7.** Additive function of actin polymerization and hydrostatic pressure in driving endothelial cell (EC) migration and sprouting angiogenesis.
(**A–D**) Representative maximum intensity projection confocal z-stacks of 28 hpf wildtype (**A, B**) and *aqp1a.1*[rk28/rk28]*;aqp8a.1*[rk29/rk29] (**C, D**) embryos treated with 0.1% DMSO (**A, C**) or 0.8 µg/ml Lat. B (**B, D**) from 20 hpf to 28 hpf (for each condition: wildtype, n = 10 embryos; *aqp1a.1*[rk28/rk28]*;aqp8a.1*[rk29/rk29], n = 12 embryos, two independent experiments). Scale bar, 20 µm. (**E**) Intersegmental vessel (ISV) length in 28 hpf wildtype and *aqp1a.1*[rk28/rk28]*;aqp8a.1*[rk29/rk29] embryos treated with 0.1% DMSO or 0.8 µg/ml Lat. B (wildtype: n = 60 vessels from 10 control embryos, n = 66 vessels from 10 Lat. B-treated embryos; *aqp1a.1*[rk28]*;aqp8a.1*[rk29]: n = 126 vessels from 12 control embryos, n = 112 vessels from 11 Lat. B-treated embryos, two independent experiments). Statistical significance was determined by Brown–Forsythe ANOVA test with Sidak's multiple-comparisons test; ns, p>0.05, **p<0.01, and ****p<0.0001. (**F**) Model of endothelial tip cell migration. Tip cells generate membrane protrusions by actin polymerization and increased hydrostatic pressure via Aquaporin-mediated water inflow. In the absence of actin polymerization and presence of Aquaporins, hydrostatic pressure deforms membranes to generate membrane protrusions. When Aquaporin function is lost, membrane protrusions is generated by actin polymerization. Fewer membrane protrusions are formed when only one mechanism is utilized, resulting in slower EC migration.

The online version of this article includes the following video and source data for figure 7:

*Figure 7 continued on next page*

*Figure 7 continued*

**Source data 1.** Raw data used to generate panel E.

**Figure 7—video 1.** Additive function of actin polymerization and hydrostatic pressure in driving EC migration and sprouting angiogenesis.

https://elifesciences.org/articles/98612/figures#fig7video1

additive functions of actin polymerization and water influx in driving EC migration during sprouting angiogenesis (*Figure 7F*).

## Discussion

In this study, we demonstrate that endothelial tip cells utilize two mechanisms of migration concurrently – actin polymerization and hydrostatic pressure – during sprouting angiogenesis in the zebrafish. The increase in hydrostatic pressure in endothelial tip cells is generated by Aqp1a.1- and Aqp8a.1-mediated water inflow. By performing detailed expression analyses, we showed that *aqp1a.1* and *aqp8a.1* are differentially expressed in newly formed vascular sprouts. *Aqp1a.1* is expressed earlier in the DA than *aqp8a.1* during development and in ECs destined to be tip cells. In newly formed ISVs, endothelial tip cells express higher *aqp1a.1* mRNA levels while stalk cells express higher *aqp8a.1* mRNA levels. In zebrafish harboring mutations in both *aqp1a.1* and *aqp8a.1*, there is delayed tip cell protrusion from the DA, decreased tip cell migration, and increased occurrence of truncated ISVs at 3 dpf. We further discovered that *aqp1a.1* and *aqp8a.1* expression is upregulated by VEGFR2 activity, suggesting that VEGFA-VEGFR2 signaling employs Aquaporin function as one mechanism to drive sprouting angiogenesis.

Several lines of evidence implicate Aquaporins in facilitating water flow into endothelial tip cells during sprouting angiogenesis. First, time-lapse imaging showed polarized localization of Aqp1a.1 and Aqp8a.1 proteins at the leading edge of migrating tip cells, suggesting that water flow occurs locally at the migrating front of the cell. Secondly, single-cell analyses revealed Aqp1a.1- and Aqp8a.1-deficient tip cells have reduced cell volume, impaired expansion of cell membranes at the leading edge, and decreased generation of stable cell protrusions. These observations implicate water flow into tip cells, leading to increased cell volume. This is confirmed by the overexpression of Aqp1a.1 in tip cells that resulted in increased cell volume. Due to the incompressible property of water within an enclosed compartment such as the cell, water influx will lead to elevation in hydrostatic pressure. At the leading edge of tip cells, increased hydrostatic pressure would expand and deform membranes to generate protrusions. Additionally, water influx can reduce cytoplasmic viscosity and increase spacing between the plasma membrane and the underlying actin cytoskeleton, promoting actin monomer diffusion, actin polymerization, and the generation of membrane protrusions to facilitate cell migration (*de Boer et al., 2023*; *Loitto et al., 2009*).

A key finding in this study is the demonstration that endothelial tip cells employ hydrostatic pressure as a second mechanism of cell migration in vivo. In a previous study, we discovered that ECs continue to migrate when actin polymerization is inhibited and in the absence of filopodia (*Phng et al., 2013*). Here, we show that hydrostatic pressure can generate force for EC migration in the absence of actin-based force generation. When both mechanisms are perturbed, there is a greater impairment in EC migration (*Figure 7D and E*). This finding highlights the importance of hydrostatic pressure as an additional mechanism that reinforces actin-based cell shape changes and motility to ensure robust migration and formation of new blood vessels in complex three-dimensional environments. Such dual mode migration is advantageous when blood vessels develop in physically confined spaces such as in the zebrafish trunk, where ECs must emerge from the DA and migrate between somite boundaries and over tissues such as the notochord and neural tube to form ISVs. Such dependence on water flow in cell migration under confined microenvironment has been reported in human MDA-MB-231 breast cancer cells and mouse S180 sarcoma cells, which show cell migration persistence in narrow channels after inhibition of actin polymerization or myosin II-mediated contractility (*Stroka et al., 2014*). In this context, AQP5 mediates water flow and the decrease or increase in AQP5 expression suppresses or enhances, respectively, cancer cell migration (*Chae et al., 2008*; *Jung et al., 2011*; *Stroka et al., 2014*).

Our findings on the endothelial function of zebrafish Aqp1a.1, which is homologous to mammalian AQP1, corroborate previous studies demonstrating a function of AQP1 in promoting cell migration. In

tumor-bearing AQP1-null mice, there is reduced tumor angiogenesis and AQP1-deficient ECs display reduced migration in an in vitro wound-healing assay (*Maltaneri et al., 2020*; *Saadoun et al., 2005*). In chick, AQP1 expression levels affect neural crest cell migration speed and direction by regulating filopodia length and stability (*McLennan et al., 2020*). The regulation of membrane protrusions has also been demonstrated for other Aquaporin proteins. For example, AQP9 weakens membrane-cytoskeleton anchorage and promote the formation of membrane protrusions such as filopodia and blebs (*Karlsson et al., 2013*; *Loitto et al., 2007*). We also demonstrate for the first time a role of Aqp8a.1 in promoting EC migration and sprouting angiogenesis in the zebrafish. However, its mammalian ortholog, AQP8, was shown to inhibit colorectal cancer cell lines in vitro (*Wu et al., 2017*), suggesting cell- and context-dependent function of AQP8 in the regulation of cell migration. The role of Aqp1a.1 and Aqp8a.1 has also been recently investigated in the zebrafish. However, the authors did not report defects in EC migration but instead observed altered ISV diameter in Aqp1a.1 and Aqp8a.1 knockout and overexpression experiments (*Chen et al., 2024*). While we corroborate a decrease in the diameter of aISVs and vISVs in *aqp1a.1*$^{rk28/rk28}$ and *aqp1a.1*$^{rk28/rk28}$;*aqp8a.1*$^{rk29/rk29}$ zebrafish, we observed a slight increase in the diameter in *aqp8a.1*$^{rk29/rk29}$ zebrafish at 2 dpf (*Figure 3—figure supplement 7A*). Furthermore, unlike described in Chen et al., we could not detect a difference in ISV diameter between control and Aqp1a.1- or Aqp8a.1-overexpressing ECs (*Figure 3—figure supplement 7B*). The disparity in phenotypes observed between our groups is likely a result of differences in the *aqp1a.1* and *aqp8a.1* mutations and the level of Aqp1a.1 or Aqp8a. overexpression generated in our respective zebrafish lines.

A central determinant of water movement is the osmotic gradient across the membrane, with water flowing in the direction of higher osmotic concentration to build up hydrostatic pressure. Given a crucial role of water flow in regulating cell volume, shape, and migration, it is also important to understand the function and distribution of ion channels, exchangers, and transporters and water channels that generate an osmotic gradient across the cell. This is reflected in a growing number of studies demonstrating a role of ion transporters in cell migration. In neutrophils, the increase in cell volume and potentiation of cell migration depend on the sodium-proton exchanger 1 (NHE1) and the chloride-bicarbonate exchanger 2 (AE2) (*Nagy et al., 2023*). In T cells, chemokine-induced migration depends on ion influx via SLC12A1, and water influx via AQP3 (*de Boer et al., 2023*). In tumor cells migrating in narrow channels, the establishment of a polarized distribution of $Na^+/H^+$ pumps and Aquaporin proteins in cell membranes is required for net inflow of ions and water at the leading edge and net outflow at the trailing edge (*Stroka et al., 2014*). Although our study identifies a function of Aqp1a.1 and Aqp8a.1 in facilitating water inflow, it remains unknown which ion channels and transporters control ion flux across the cell membrane to modulate water flow in tip cells. Results from our study suggest a role of SWELL1 in generating an osmotic gradient that controls EC migration since its inhibition resulted in reduced membrane protrusions, slower tip cell migration, and defective sprouting angiogenesis. Due to the limitations of our experimental approach, we do not know whether the effect of inhibiting SWELL1 function is cell autonomous or a consequence of altered chloride flux in the neighboring cells that would alter tissue osmolarity. It also remains to be clarified where SWELL1 is localized within migrating ECs, how it regulates the osmotic gradient within the EC (cellular) and between the EC and the surrounding interstitial tissue (extracellular) to control the direction of water flow, and whether other ion channels also work in concert to modulate overall cellular and extracellular osmolarity to direct EC migration.

We have also observed enrichment of Aqp1a.1 and Aqp8a.1 proteins in specific compartments of ECs at the leading edge of migrating endothelial tip cells as well as apical membranes during lumen formation (not shown). This leads to the question of how Aquaporin localization is regulated as this will determine the site of increased water inflow and hydrostatic pressure. It has been shown that rapid changes in subcellular localization of mammalian Aquaporins upon stimulation occurs mainly via trafficking to and from the plasma membrane (*Markou et al., 2022*) to modulate the amount of protein in the plasma membrane. The regulation of AQP subcellular relocalization via calmodulin- and/ or phosphorylation-dependent mechanisms has been implicated for AQP0-5 and AQP7-9 (*Markou et al., 2022*). For example, rapid translocation of AQP1 to plasma membrane upon a hypotonic stimulus is dependent on calmodulin activation and phosphorylation by protein kinases C (PKC) (*Conner et al., 2012*) and subcellular localization of AQP8 in hepatocytes is also regulated by PKA and PI3K signaling (*Gradilone et al., 2005*; *Gradilone et al., 2003*). Further work is needed to elucidate the mechanism of Aqp1a.1 and Aqp8a.1 protein distribution in ECs.

In this study, we have assumed that Aqp1a.1 and Aqp8a.1 regulate EC behaviors through their ability to control water permeation based on previous studies demonstrating that water molecules flow through mammalian AQP1 (*Preston et al., 1992*; *Zeidel et al., 1992*) and AQP8 (*Liu et al., 2006*; *Ma et al., 1997*), as well as through zebrafish orthologs Aqp1a.1 and Aqp8a.1 (*Tingaud-Sequeira et al., 2010*). However, AQP1 and AQP8 are not water-specific channels. AQP1 can conduct unpolar gases such as carbon dioxide (*Nakhoul et al., 1998*; *Prasad et al., 1998*; *Wang et al., 2007*) and nitric oxide (*Herrera et al., 2006*; *Herrera and Garvin, 2007*), and in zebrafish, Aqp1a.1 also mediates the transfer for small gaseous molecules such as carbon dioxide and ammonia (*Talbot et al., 2015*). AQP8 is permeable to ammonia (*Jahn et al., 2004*; *Liu et al., 2006*) and is present in the inner mitochondrial membrane (*Calamita et al., 2005*), where it is suggested to mediate ammonia transport rather than water fluxes (*Soria et al., 2010*). Nevertheless, because water movement flows up an osmotic gradient, increased entry of ions into the cell through Aquaporins will also trigger water inflow.

In summary, our study highlights the role of water influx and hydrostatic pressure as another force-generating mechanism utilized by cells to build tissues during animal development. We demonstrate that endothelial tip cells employ both actin polymerization and hydrostatic pressure for robust sprouting angiogenesis. As morphogenetic events are governed by a coordination of cell shape changes, cell migration, and rearrangements, we envision that the shaping and formation of other tissues will also depend on Aquaporin function and changes in hydrostatic pressure.

# Materials and methods

**Key resources table**

| Reagent type (species) or resource | Designation | Source or reference | Identifiers | Additional information |
|---|---|---|---|---|
| Strain, strain background (*Danio rerio*) | AB | Zebrafish International Resource Center | RRID:ZIRC_ZL1 | |
| Strain, strain background (*Danio rerio*) | Tg(kdrl:EGFP)[s843] | *Jin et al., 2005* | ZFIN:ZDB-ALT-050916-14 | |
| Strain, strain background (*Danio rerio*) | Tg(fli1:zgc:114046-EGFP)[ncv69] | *Ando et al., 2019* | ZFIN:ZDB-ALT-200226-15 | Referred to as *Tg(fli1a:H2B-EGFP)[ncv69]* |
| Strain, strain background (*D. rerio*) | Tg(fli1:Lifeact-mCherry)[ncv7] | *Wakayama et al., 2015* | ZFIN:ZDB-ALT-150603–1 | |
| Strain, strain background (*Danio rerio*) | Tg(fli1:myr-mCherry)[ncv1] | *Fukuhara et al., 2014* | ZFIN:ZDB-ALT-131216–1 | |
| Strain, strain background (*Danio rerio*) | Tg(fli1:EGFP-Hsa.PLCD1)[rk26] | *Kondrychyn et al., 2020* | ZFIN:ZDB-ALT-201118-4 | |
| Strain, strain background (*Danio rerio*) | Tg(kdrl:Hsa.HRAS-mCherry)[s916] | *Hogan et al., 2009* | ZFIN:ZDB-ALT-090506-2 | |
| Strain, strain background (*Danio rerio*) | "Tg(kdrl:nls-EGFP)[ubs1];(fli1:Hsa.B4GALT1-mCherry)[bns9]" | *Kwon et al., 2016* | ZFIN:ZDB-ALT-081105–1; ZFIN:ZDB-ALT-160804–2 | |
| Strain, strain background (*Danio rerio*) | Tg(fli1:GAL4FF)[ubs3] | *Herwig et al., 2011* | ZFIN:ZDB-ALT-120113-6 | |
| Strain, strain background (*Danio rerio*) | Tg(fli1ep:aqp1a.1-mEmerald)[rk30] | This study | | See Materials and methods 'Plasmid construction and transgenesis' |
| Strain, strain background (*Danio rerio*) | Tg(fli1ep:aqp8a.1-mEmerald)[rk31] | This study | | See Materials and methods 'Plasmid construction and transgenesis' |

*Continued on next page*

*Continued*

| Reagent type (species) or resource | Designation | Source or reference | Identifiers | Additional information |
|---|---|---|---|---|
| Cell line (*Homo sapiens*) | HAEC-Human Aortic Endothelial Cells | Lonza | Lonza:CC-2535 | lot-20TL231227 |
| Antibody | Anti-digoxigenin (sheep polyclonal) Fab fragments, conjugated with alkaline phosphatase | Roche | Roche:11093274910; RRID:AB_514497 | Whole-mount in situ hybridization (1:5000) |
| Recombinant DNA reagent | *mEmerald-N1* (plasmid) | Addgene | RRID:Addgene_53976 | A source of mEmerald protein for In-Fusion cloning |
| Recombinant DNA reagent | *Ds-MCS* (plasmid) | *Emelyanov et al., 2006* | N/A | The minimal *cis*-required sequences of *Dissociation (Ds)* element from the maize (used in In-Fusion cloning) |
| Recombinant DNA reagent | *NLS-AcTP-SP6* (plasmid) | *Emelyanov et al., 2006* | N/A | *Activator (Ac)* transposase cDNA from the maize |
| Recombinant DNA reagent | *Ds-fli1ep:NLS-EGFP-P2A-mKate2-GM130* (plasmid) | This study | | In-Fusion cloning |
| Recombinant DNA reagent | *Ds-6xUAS:aqp1a.1-P2A-EGFP-T2A-mKate2-CAAX* (plasmid) | This study | | In-Fusion cloning |
| Recombinant DNA reagent | *Ds-kdrl:mEmerald* (plasmid) | This study | | In-Fusion cloning |
| Sequence-based reagent | RNAscope Probe-*Dr-aqp1a.1* | Advanced Cell Diagnostics | ADC:893521 | |
| Sequence-based reagent | RNAscope Probe-*Dr-aqp8a.1-C3* | Advanced Cell Diagnostics | ADC:802961-C3 | |
| Sequence-based reagent | RNAscope Probe-*Dr-lrrc8aa-C2* | Advanced Cell Diagnostics | ADC:1596611-C2 | |
| Software, algorithm | Huygens Essential 22.10 | Scientific Volume Imaging B.V. | RRID:SCR_014237 | https://svi.nl/Huygens-Essential |
| Software, algorithm | Fiji | *Schindelin et al., 2012* | RRID:SCR_002285 | https://fiji.sc |
| Software, algorithm | Cell Ranger v2.1 | 10X Genomics | RRID:SCR_017344 | https://www.10xgenomics.com/support/software/cell-ranger |
| Software, algorithm | Seurat package v4.3.0 | *Hao et al., 2021* | RRID:SCR_016341 | https://satijalab.org/seurat |
| Software, algorithm | ShinyCell package v2.1 | *Ouyang et al., 2021* | RRID:SCR_022756 | https://github.com/SGDDNB/ShinyCell |
| Software, algorithm | Prism v10.2.0 | GraphPad | RRID:SCR_002798 | https://www.graphpad.com |
| Software, algorithm | QuantStudio Design and Analysis software v1.5.3 | Applied Biosystems | N/A | https://www.thermofisher.com |

## Zebrafish maintenance and stocks

Zebrafish (*Danio rerio*) were raised and staged according to established protocols (*Kimmel et al., 1995*). Transgenic lines used in this work are *Tg(kdrl:EGFP)*$^{s843}$ (*Jin et al., 2005*), *Tg(fli1:zgc:114046-EGFP)*$^{ncv69}$ (referred to as *Tg(fli1a:H2B-EGFP)*$^{ncv69}$, *Ando et al., 2019*), *Tg(fli1:Lifeact-mCherry)*$^{ncv7}$ (*Wakayama et al., 2015*), *Tg(fli1:myr-mCherry)*$^{ncv1}$ (*Fukuhara et al., 2014*), *Tg(fli1:EGFP-Hsa.PLCD1)*$^{rk26}$ (*Kondrychyn et al., 2020*), *Tg(kdrl:Hsa.HRAS-mCherry)*$^{s916}$ (*Hogan et al., 2009*), *Tg(kdrl:nls-EGFP)*$^{ubs1}$;*(fli1:Hsa.B4GALT1-mCherry)*$^{bns9}$ (*Kwon et al., 2016*), and *Tg(fli1:GAL4FF)*$^{ubs3}$ (*Herwig et al., 2011*). Zebrafish were maintained on a 14 hr light/10 hr dark cycle, and fertilized eggs were collected and raised in E3 medium at 28°C. To inhibit pigmentation in embryos older than 24 hpf, 0.003% *N*-Phenylthiourea (Sigma-Aldrich, Cat# P7629) in E3 medium was used. All animal experiments were approved by the Institutional Animal Care and Use Committee at RIKEN Kobe Branch (IACUC).

## Plasmid construction and transgenesis

Plasmid *mEmerald-N1* was a gift from Michael Davidson (Addgene plasmid #53976), plasmids *pDs-MCS* containing the minimal *cis*-required sequences of *Dissociation (Ds)* transposable element from the maize and *pNLS-AcTP-SP6* containing an ORF of *Activator (Ac)* transposase from the maize were a gift from Sergei Parinov. Other plasmids used in this study were generated by using In-Fusion HD Cloning kit (Takara Bio Inc, Cat# 639650). Detailed information regarding plasmids used in this study can be found in *Supplementary file 1*. Tg(fli1ep:aqp1a.1-mEmerald)[rk30] and Tg(fli1ep:aqp8a.1-mEmerald)[rk31] transgenic zebrafish were generated by injecting *Tol2*-based *fli1ep:aqp1a.1-mEmerald* or *fli1ep:aqp8a.1-mEmerald* plasmids (10 ng/µl), respectively, along with *Tol2 transposase* mRNA (100 ng/µl) into one-cell stage AB embryos. *Tol2* transposase mRNA was transcribed from NotI-linearized *pCS-TP* plasmid (a gift from Koichi Kawakami, National Institute of Genetics, Japan) using the mMESSAGE mMACHINE SP6 kit (Invitrogen, Cat# AM1340). Injected embryos were raised to adult and screened for founders.

## Cloning of *aqp1a.1* and *aqp8a.1* genes

Total RNA was isolated from 1-day-old zebrafish embryos with TRI Reagent using Direct-zol RNA MicroPrep kit (Zymo Research, Cat# R2061-A) according to the manufacturer's protocol. The first-strand cDNA was synthesized from 1 µg of a total RNA by oligo(dT) priming using the SuperScript III First-Strand synthesis system (Invitrogen, Cat# 18080-051) according to the manufacturer's protocol. Amplification of cDNAs was performed using a high-fidelity KOD-Plus-Neo DNA polymerase (Toyobo, Japan, Cat# KOD-401) and resulting PCR products were cloned using NEB PCR Cloning kit (New England BioLabs, Cat# E1202S). Positive clones and plasmids were verified by DNA sequencing.

## Mosaic expression of DNA constructs

To avoid unwanted *Tol2* re-transposition from the donor site in transgenic zebrafish lines generated using *Tol2* transposon (*Kondrychyn et al., 2009*), *Ac/Ds* transposon system from the maize (*Emelyanov et al., 2006*) was used for mosaic expression of DNA constructs. *Ds*-based plasmids (10 pg) were co-injected with 100 pg of *Ac* transposase mRNA, transcribed from BamHI-linearized *pNLS-AcTP-SP6* plasmid using the mMESSAGE mMACHINE SP6 kit (Invitrogen), into one-cell stage zebrafish wildtype or *aquaporin* mutant embryos. Embryos were analyzed at 25–26 hpf after injections.

## In vivo cell volume analysis

Single-cell labeling of ECs in ISVs was achieved by mosaic expression of *pDs-kdrl:mEmerald* plasmid in wildtype or *aqp1a.1[rk28/rk28];aqp8a.1[rk29/rk29]* mutant *Tg(fli1:Lifeact-mCherry)[ncv7]* transgenic embryos. To visualize single ECs with Aqp1a.1 overexpression, we injected *pDs-6xUAS:aqp1a.1-P2A-EGFP-T2A-mKate2-CAAX* plasmid into *Tg(fli1:GAL4FF)[ubs3];Tg(fli1:myr-mCherry)[ncv1]* embryos. Embryos were imaged at 25–26 hpf with an Olympus UPLSAPO ×60/NA 1.2 water immersion objective with optical Z planes interval of 0.26 µm. Cell volume was measured with Huygens Essential 22.10 software (Scientific Volume Imaging B.V.) using object analyzer tool with manual adjustment of threshold.

## Generation of *aqp1a.1* and *aqp8a.1* zebrafish mutants

Aquaporins mutants were generated by CRISPR/Cas9-mediated mutagenesis. Guide RNAs (gRNAs) targeting the first exons of *aqp1a.1* and *aqp8a.1* (see *Figure 3—figure supplements 1 and 2*) were generated by using cloning-independent protocol (*Gagnon et al., 2014*). Briefly, to generate templates for gRNA transcription, a 60-base oligodeoxynucleotides containing the SP6 promoter sequence, the gene-specific sequence (*aqp1a.1*: 5'-GACAGCTGGCCAGCAGACCC-3'; *aqp8a.1*: 5'-GATGTCTCCCCCATCGCCCG-3') and 23-base overlap region was annealed to an 80-base constant oligodeoxynucleotide encoding the reverse-complement of the tracrRNA tail (see *Supplementary file 2* for sequence information). The ssDNA overhangs were filled-in with T4 DNA polymerase (New England BioLabs, Cat# M0203S) to generate dsDNA. The gRNAs were in vitro transcribed using MEGAScript SP6 kit (Invitrogen, Cat# AM1330), DNase treated and purified with RNA Clean and Concentrator kit (Zymo Research, Cat# R1015). The gRNAs were quality-checked by running 1 µg of the product on a 2% TBE-agarose gel. Cas9/gRNA RNP complex was assembled just prior to injection and after 5 min incubation at room temperature 200 pg of Cas9 protein (Invitrogen, Cat# A36497) and 200 pg of gRNAs (100 pg of each, *aqp1a.1* gRNA and *aqp8a.1* gRNA) were co-injected into one-cell

stage *Tg(fli1:myr-mCherry)*[ncv1] embryos. Such dose produced over 70% embryos survival post-injection showed CRISPR/Cas9-induced somatic *aqp1a.1* and *aqp8a.1* gene mutations. $F_0$ founders were identified by outcrossing CRISPR/Cas9-injected fish with wildtype fish and screening the offspring for mutations at 2 dpf using Sanger sequencing. We found one $F_0$ fish with germline transmitted mutations in both *aqp1a.1* and *aqp8a.1* genes and outcrossed it with *Tg(fli1:myr-mCherry)*[ncv1] and *Tg(fli1a:H2B-EGFP)*[ncv69] transgenic fish to establish $F_1$ generation of heterozygotes. A 55 $F_1$ fish were genotyped to identify 4 *aqp1a.1*[+/rk28], 11 *aqp8a.1*[+/rk29], and 12 *aqp1a.1*[+/rk28];*aqp8a.1*[+/rk29] heterozygote fish. Subsequently, $F_1$ heterozygote fish were in-crossed to establish a single and double homozygote fish.

## DNA isolation and genotyping

Genomic DNA was isolated from either embryos or fin clips using HotSHOT method (*Meeker et al., 2007*). To identify genomic lesions, the following primers were used: aqp1a1-fwd (5′-CGCCTCCAGATTCATTAGCAGGA-3′) and aqp1a1-rev (5′-GTAAGTGAACTGCTGCCAGTGA-3′) to amplify a 550 bp fragment, and aqp8a1-fwd (5-GGATCAATTGAGTTGCATAACAGAC-3′) and aqp8a1-rev (5′-CTGTAATGTAGACTTGTAAAGTGGA-3′) to amplify a 638 bp fragment. Mutations were assessed by direct sequencing of purified PCR products.

## Quantitative real-time PCR

Total RNA from whole embryos was isolated with TRI reagent using Direct-zol RNA MicroPrep kit (Zymo Research) according to the manufacturer's protocol, including on-column DNA digestion. RNA was quantified using a NanoDrop 1000 spectrophotometer (Thermo Fisher Scientific, USA). cDNA was synthesized from 300 ng of purified RNA using LunaScript RT SuperMix kit (New England BioLabs, Cat# E3030L) according to the manufacturer's protocol. Amplification of target cDNA was performed in technical triplicate of three biological replicates using the SYBR green methods. Each qPCR reaction mixture contained 5 µl 2× Luna Universal qPCR master mix (New England BioLabs, Cat# M3003L), 1 µl cDNA (twofold dilution), 0.2 µl antarctic thermolabile UDG (New England BioLabs, Cat# M0372L), and 250 nM each primer to a final volume of 10 µl. Reactions were run in 384-well plates (Applied Biosystems, USA) using the QuantStudio 5 Real-Time PCR system (Applied Biosystems) with the following thermal cycling conditions: initial UDG treatment at 25°C for 2 min, UDG inactivation at 50°C for 5 min, and denaturation at 95°C for 1 min, followed by 45 cycles of 15 s at 95°C, 30 s at 60°C with a plate read at the end of the extension step. Control reactions included a no-template control (NTC) and a no-reverse transcriptase control (NRT). Dissociation analysis of the PCR products was performed by running a gradient from 60 to 95°C to confirm the presence of a single PCR product. Amplification data was analyzed using QuantStudio Design and Analysis software v1.5.3 (Applied Biosystems), and relative fold change was calculated using Pfaffl method (*Pfaffl, 2001*) and normalized to *gapdh* expression (as an internal control). Primers are listed in *Supplementary file 2*. Two embryos were used in each biological replicate to generate an average value that was used to calculate the final mean ± SD from two or three independent experiments.

## Whole-mount RNA in situ hybridization

### RNAscope in situ hybridization

RNAscope in situ hybridization was conducted by using RNAscope Multiplex Fluorescent Reagent kit v2 (Advanced Cell Diagnostics, Cat# 323100). We adapted manufacture's protocol designed for samples mounted on slides and protocol developed for whole-mount embryo samples (*Gross-Thebing et al., 2014*). For each experimental point, five embryos were processed in one 1.5 ml Eppendorf tube. Briefly, 20 hpf and 22 hpf old embryos were manually dechorionated and fixed in freshly prepared 4% methanol-free PFA (Thermo Fisher Scientific, Cat# 28908) in PBS for 1 hr at room temperature. After fixation, embryos were washed three times in PBS containing 0.01% Tween-20 (PBST), dehydrated stepwise in 5 min washes in a series of increasing methanol concentrations (25%, 50%, 75%, 100%) in PBS, and then stored in 100% methanol at –20°C before use them for hybridization. The methanol-stored embryos were incubated with 5% $H_2O_2$ in methanol for 20 min at room temperature and then rehydrated stepwise in 5 min washes in series of decreasing methanol concentrations (75%, 50%, 25%) in PBS, followed by three times 5 min washes in PBST. Embryos were permeabilized in 2 drops of RNAscope Protease III from the kit for 15 min, rinsed three times with PBST, and hybridized with RNAscope target probes overnight at 50°C in water bath. After hybridization probes

were recovered and embryos were washed three times with 0.2× SSCT (0.01% Tween-20 in SSC) at room temperature, re-fixed with 4% PFA for 10 min, and washed three times with 0.2× SSCT. For RNA detection, the embryos were sequentially hybridized with three different amplifier solutions for 30 min (Amp1 and Amp2) and 15 min (Amp3) in a water bath at 40°C. After each hybridization step, the embryos were washed three times with 0.2× SSCT for 10 min at room temperature. To develop signal of each probe, the embryos were sequentially incubated in a water bath at 40°C with (i) the horseradish peroxidase (HRP) for 15 min, (ii) TSA fluorophore for 30 min, and (iii) the HRP blocker for 15 min. After each incubation step, the embryos were washed three times with 0.2× SSCT for 10 min at room temperature. HRP step is linked to a probe channel, we first developed C1 probe, *Dr-aqp1a.1* (Advanced Cell Diagnostics) using HRP-C1 and TSA Vivid Fluorophore 520 (dilution 1:1500, Tocris, Cat# 7523), then C2 probe, *Dr-lrrc8aa* (Advance Cell Diagnostics) using HRP-C2 and TSA Vivid Fluorophore 570 (dilution 1:1500, Tocris, Cat# 7526), and finally C3 probe, *Dr-aqp8a.1* (Advanced Cell Diagnostics) using HRP-C3 and TSA Vivid Fluorophore 650 (dilution 1:1500, Tocris, Cat# 7527). Nuclei were counterstained with DAPI ready-to-use solution overnight at 4°C in the dark. Prior to imaging embryos were rinsed in PBST and kept in 70% glycerol in PBS at 4°C in the dark. Images were acquired using Olympus FV3000 confocal microscope and an Olympus UPlanXApo ×60/NA 1.42 oil immersion objective.

## Chromogenic in situ hybridization

Chromogenic in situ hybridization was conducted according to standard protocol (*Thisse and Thisse, 2008*) with minor modifications. For each experimental point, 10 embryos were processed in one 1.5 ml Eppendorf tube. Briefly, 30 hpf, 48 hpf, 72 hpf, and 96 hpf old embryos were manually dechorionated and fixed in freshly prepared 4% methanol-free PFA in PBS (pH 7.4) at 4°C overnight. After fixation, embryos were washed three times in PBS containing 0.1% Tween-20 (PBST), dehydrated stepwise in 5 min washes in a series of increasing methanol concentrations (25%, 50%, 75%, 100%) in PBS and then stored in 100% methanol at –20°C before use them for hybridization. The methanol-stored embryos were rehydrated stepwise in 5 min washes in series of decreasing methanol concentrations (75%, 50%, 25%) in PBS followed by three times 5 min washes in PBST, and permeabilized 20 min with 10 µg/ml proteinase K (Thermo Fisher Scientific, Cat# EO0491). Hybridization was carried out in buffer (50% deionized formamide, 5× SSC, 50 µg/ml heparin, 500 µg/ml tRNA [Invitrogen, Cat# 15401029], 10 mM citric acid and 0.1% Tween-20) containing 5% dextran sulfate (500 kDa, Wako Chemicals, Japan, Cat# 193-09981) at 69°C overnight. After stringency wash, the specimens were blocked with 2% blocking reagent (Roche, Cat# 11096176001) in maleic acid buffer (100 mM maleic acid, 150 mM NaCl, 50 mM $MgCl_2$ and 0.1% Tween-20, pH 7.5) and incubated overnight with anti-digoxigenin antibody, conjugated with alkaline phosphatase (dilution 1:5000, Roche, Cat# 11093274910) at 4°C. For detection of alkaline phosphatase, specimens were incubated in staining buffer (50 mM Tris-HCl, 50 mM NaCl, 25 mM $MgCl_2$, 2% polyvinyl alcohol, and 0.1% Tween-20, pH 9.5) containing 375 µg/ml nitro-blue tetrazolium chloride (Roche, Cat# 11383213001) and 175 µg/ml 5-bromo-4-chloro-3′-indolyl-phosphate (Roche, Cat# 11383221001). Embryos were cleared in 70% glycerol overnight and imaged using Leica M205FA microscope.

## RNA probe synthesis

The cDNA-containing vectors were linearized with appropriate restriction enzymes (detailed maps of vectors will be provided upon request) and used as a template for RNA probe synthesis. Sense and antisense RNA probes were synthesized using MEGAscript SP6 (Invitrogen) or T7 (Invitrogen, Cat# AM1333) kits and digoxigenin (DIG)-labeled rNTPs (Roche, Cat# 11277073910) according to the manufacturer's protocol and purified using RNA Clean and Concentrator kit (Zymo Research). The following sense and antisense DIG-labeled riboprobes were generated: (i) *aqp1a.1*, 1.1 kb long probe comprised 121 nt of 5′UTR, 783 nt of the open-reading frame (ORF) and 207 nt of 3′UTR; and (ii) *aqp8a.1*, 0.8 kb long probe comprised 17 nt of 5′UTR, 783 nt of ORF and 30 nt of 3′UTR. Sense probes showed no specific or unspecific staining (data not shown).

## Image processing

Chromogenic and RNAscope in situ hybridization images were processed using Fiji software with brightness and contrast adjustments. Z-stacks of fluorescent images are presented as maximum intensity projections.

## Analysis of mRNA expression in tip and stalk cells

RNAscope processed embryos were mounted on slide and imaged using Olympus FV3000 confocal microscope and Olympus UPlanXApo ×60/NA 1.42 oil immersion objective. Maximum intensity projection of z-stacks was used to determine the level of cellular fluorescence in Fiji. Tip and stalk cells were selected using freehand selection tool; 'mean grey value', 'area', and 'integrated density' in each ROI were measured first for channel 1 (aqp1a.1). A region next to cell without fluorescence was selected as background and measured. The step was repeated for channel 2 (aqp8a.1) on the same ROIs. Two parameters were calculated for each channel, (i) the corrected total cell fluorescence, CTCF = Integrated Density – (Area of selected cell × Mean fluorescence of background) and (ii) ratio, $R$ = CTCF of tip cell/CTCF of stalk cell. We consider that if $R < 0.9$, expression is lower in tip cell, if $R$ = 0.9–1.1, expression is equal in tip and stalk cells, if $R > 1.1$, expression is higher in tip cell.

## Imaging

For live confocal imaging, embryos were mounted in 0.8% low-melt agarose (Bio-Rad, Cat# 1613111) in E3 medium containing 0.16 mg/ml Tricaine (Sigma-Aldrich, Cat# E10521-10G) and 0.003% phenylthiourea (Sigma-Aldrich) in glass-bottom 35 mm dishes (MatTek, USA). Confocal z-stacks were acquired using an inverted Olympus IX83/Yokogawa CSU-W1 spinning disc confocal microscope equipped with a Zyla 4.2 CMOS camera (Andor) and Olympus UPLSAPO ×40/NA 1.25 or ×30/NA 1.05 silicone oil immersion objectives. Bright-field images were acquired on Leica M205FA microscope. Images were processed using Fiji software.

## Analysis of cell migration

aqp1a.1$^{+/rk28}$;aqp8a.1$^{+/rk29}$ and aqp1a.1$^{rk28/rk28}$;aqp8a.1$^{rk29/rk29}$ embryos on Tg(fli1a:H2B-EGFP)$^{ncv69}$;(fli1:Lifeact-mCherry)$^{ncv7}$ double transgenic background were imaged from 21 hpf to 30 hpf with time interval 6 min using Olympus UPLSAPO ×30/NA 1.05 silicon oil immersion objective. Both heterozygote and mutant embryos were mounted on the same dish (1 het plus 2 mut, or 2 het plus 2 mut) to image embryos under the same conditions. Acquired time-lapse images were registered using HyperStackReg plugin in Fiji. Tip cell nuclei and leading edge were tracked using Manual Tracking plugin and velocity was calculated.

## Analysis of grow rate of actin bundles in filopodia

Wildtype and aqp1a.1$^{rk28/rk28}$;aqp8a.1$^{rk29/rk29}$ embryos on Tg(fli1:EGFP-Hsa.PLCD1)$^{rk26}$;(fli1:Lifeact-mCherry)$^{ncv7}$ double transgenic background were imaged from 25 hpf for 10 min with time interval of 30 s using Olympus UPLSAPO ×60/NA 1.2 water immersion objective. Actin bundles growth was tracked using Manual Tracking plugin in Fiji and velocity was calculated.

## Analysis of filopodia number and length

Wildtype, aqp1a.1$^{+/rk28}$;aqp8a.1$^{+/rk29}$ and aqp1a.1$^{rk28/rk28}$;aqp8a.1$^{rk29/rk29}$ embryos on Tg(fli1:EGFP-Hsa.PLCD1)$^{rk26}$;(fli1:Lifeact-mCherry)$^{ncv7}$ double transgenic background were imaged from 24 hpf for 15 min with a time interval of 50 s using Olympus UPLSAPO ×60/NA 1.2 water immersion objective. The number of filopodia per 100 µm membrane length was calculated at each time point by counting the number of filopodia number per vessel and measuring membrane length. The average filopodia number per 100 µm of each vessel was calculated by averaging the number of filopodia per 100 µm at each time point.

## Assessment of tip cell polarization

Tg(kdrl:nls-EGFP)$^{ubs1}$;(fli1:Hsa.B4GALT1-mCherry)$^{bns9Tg}$ wildtype and aqp8a.1$^{rk29/rk29}$ embryos were imaged at 24–26 hpf using an Olympus UPLSAPO ×40/NA 1.25 silicon oil immersion objective. To analyze endothelial tip cell polarity in aqp1a.1$^{rk28/rk29}$ embryos, Golgi apparatus and nuclei were labeled by mosaic expression of pDs-fli1ep:nls-EGFP-P2A-mKate2-GM130 plasmid in Tg(fli1-EGFP-Hsa.

*PLCD1)$^{rk26}$* transgenic line. Images were analyzed using Fiji software. To assess polarity, the nucleus was fit into an ellipsoid shape and the angle between the primary axis of the ellipse and the center of the Golgi was measured (see *Figure 4—figure supplement 1D*). Polarization of ECs is defined as follows: (1) front (polarized), if the Golgi is located within $-30^0 \sim +30^0$; (2) none, if the Golgi is located within $+30^0 \sim +150^0$ or $-30^0 \sim -150^0$; or (3) rear, if the Golgi is located on the downstream side of the nucleus and angle is $-150^0 \sim +150^0$. Tip cells in ISVs no. 5–15 were analyzed.

## Quantification of EC number and ISV diameter

Measurements were performed in wildtype, *aqp1a.1$^{rk28/rk28}$*, *aqp8a.1$^{rk29/rk29}$* and *aqp1a.1$^{rk28/rk28}$;aqp8a.1$^{rk29/rk29}$* zebrafish raised in *Tg(fli1:Lifeact-mCherry)$^{ncv7}$;(fli1a:H2B-EGFP)$^{ncv69}$* transgenic line or in *Tg(fli1ep:aqp1a.1-mEmerald)$^{rk30}$;(fli1:myr-mCherry)$^{ncv1}$* and *Tg(fli1ep:aqp8a.1-mEmerald)$^{rk31}$;(fli1:myr-mCherry)$^{ncv1}$* zebrafish. Confocal z-stacks images of embryos were taken at 50–54 hpf and 3 dpf using Olympus UPLSAPO ×40/NA 1.25 silicone oil immersion objective. ISVs no. 5–15 were used for quantification. For ISV diameter, five measurements were made along the length of each ISV using Fiji software and the average was plotted.

## Chemical treatment

Latrunculin B (Merck Millipore, Cat# 428020-1MG) was dissolved in DMSO to 1 mg/ml and stored at –20°C. Ki8751 (Selleck Chemicals, Cat# S1363) was prepared as 5 mM solution in DMSO and stored at –80°C. DCPIB (Tocris, Cat# 1540) was prepared as 10 mM solution in EtOH and stored at –20°C. All compounds were diluted to the desired concentration in E3 medium (see in figure legends). Embryos were treated from 20 hpf for 6 (Ki8751 and DCPIB) or 8 hr (Latrunculin B). For imaging experiments, the same concentrations of chemicals were added to the agarose and E3 medium.

## Cell culture

Adult HAECs (Lonza, Cat# CC-2535, lot-20TL231227) were cultured in EGM medium (Lonza, Cat# CC-3124) and used at passage 3. Cells were seeded in EBM medium (Lonzo, Cat# CC-3162) supplemented with 2% FBS (Gibco, Cat# 26240079) at $4 \times 10^5$ cells/well on 24-well plate (Corning, Cat# 3524) coated with 5 µg/ml fibronectin (Sigma-Aldrich, Cat# F0895). Cells were treated with either 0.01% DMSO or different concentrations of Ki8751 inhibitor for 6 hr. Cells were lysed with TRI reagent and RNA was isolated using Direct-zol RNA MicroPrep kit (Zymo Research).

## Single-cell RNA sequencing

ECs were isolated from *Tg(kdrl:EGFP)$^{s843}$* transgenic embryos. Cell sorting was carried out with FACSArialI Cell Sorter (BD Bioscience, USA). Single-cell suspension was loaded into the 10X Chromium system and cDNA libraries were constructed using Chromium Next GEM Single Cell 3′ GEM, Library and Gel Bead Kit v2 (10X Genomics, USA) according to manufacturer's protocol. Library was sequenced on the Illumina HiSeq 1500 Sequencer (Illumina, USA). Cell Ranger v2.1 was used to de-multiplex raw base call (BCL) files generated by Illumina sequencers into FASTQ files.

## Single-cell RNA sequencing data analysis

The raw sequence data from above mentioned internal sequencing and also a public zebrafish embryo data at 24 hpf stage (*Gurung et al., 2022*) were processed. The public dataset was obtained from NCBI GEO database (accession number GSE202912). The raw sequencing reads from both datasets were mapped to the zebrafish genome assembly (GRCz11, Ensembl release 112). Further analyses were performed using Seurat package (version 4.3.0) in R software (https://www.r-project.org). The expression matrices were first filtered by keeping genes that are expressed in a minimum of three cells and cells that expressed a minimum of 200 genes for downstream analysis. The data were then normalized using NormalizeData function that normalizes the gene expression for each cell by the total expression counts (with a scale factor 10,000). To correct for batch effect between the data from the two time points, the rpca method in the Seurat package was applied to integrate the data. The top 2000 variable genes identified using the vst method in FindVariableFeatures function were used for principal component analysis in RunPCA function, and the first 30 principal components were

used for visualization analysis with Uniform Manifold Approximation and Projection (UMAP) method, and in FindNeighbors function analysis. The cell clustering resolution was set at 0.5 in FindClusters function. The Seurat object was then processed using the ShinyCell package (version 2.1) for gene expression visualizations.

## Statistical analysis

For zebrafish experiments, the sample size (number of cells or vessels) was chosen based on the number of embryos expressing the transgene of interest (e.g., transient overexpression of plasmids) or of the desired genotype obtained per experiment. The number of embryos used for time-lapse imaging per experiment ranged from 3 to 4 per experiment for overnight imaging, 6–8 for short (0.5–2 hr) but high temporal resolution imaging, and 1–3 embryos for drug treatments since time was a limiting factor. Statistical analysis was performed using Prism software version 10.2.0 (GraphPad). The variance between the mean values of two groups was evaluated using the unpaired Student's *t*-test. For assessment of more than three groups, we used one-way ANOVA test. A p-value of $<0.05$ was considered statistically significant. Statistic details can be found in each figure legend.

## Materials availability statement

Further information and requests for reagents should be directed to the corresponding author, Li-Kun Phng (likun.phng@riken.jp).

## Acknowledgements

We thank members of the Phng Lab, Y-C. Wang, and S Thukral for discussions and suggestions; G Chen, E Taniguchi, A Nomori, J Chong, RIKEN BDR Research Aquarium, and RIKEN Kobe BioImaging Facilities & Factory for technical assistance; J Vermot, M Francois, and A Yap for comments on the manuscript. This work was supported by core funding from RIKEN BDR (to LKP), RIKEN BDR-Otsuka Pharmaceutical Collaboration Center (IK), the Naito Foundation (LKP), the JSPS Grants-in-Aid for Scientific Research (KAKENHI) grants (22H02624, 22H05168 to LKP; 22K06244 to IK), Swedish Research Council (2015-00550 to CB); Swedish Cancer Society (2018/449, 2018/1154 to CB); Knut and Alice Wallenberg Foundation (2020.0057, 2018.0218 to CB); Swedish Brain Foundation (ALZ2019-0130, ALZ2022-0005 to CB); the Leducq Foundation (22CVD01, 23CVD02 to CB) and the Innovative Medicines Initiative (IM2PACT-807015 to CB).

## Additional information

### Funding

| Funder | Grant reference number | Author |
|---|---|---|
| RIKEN Center for Biosystems Dynamics Research | Intramural funding | Li-Kun Phng |
| Naito Foundation | The Naito Grant for female scientist after maternity leave | Li-Kun Phng |
| Japan Society for the Promotion of Science | 22H02624 | Li-Kun Phng |
| Japan Society for the Promotion of Science | 22K06244 | Igor Kondrychyn |
| Vetenskapsrådet | 2015-00550 | Christer Betsholtz |
| Cancerfonden | 2018/449 | Christer Betsholtz |
| Knut and Alice Wallenberg Foundation | 2020.0057 | Christer Betsholtz |
| Swedish Brain Foundation | ALZ2019-0130 | Christer Betsholtz |

| Funder | Grant reference number | Author |
|---|---|---|
| Leducq Foundation | 22CVD01 | Christer Betsholtz |
| Innovative Medicines Initiative | IM2PACT-807015 | Christer Betsholtz |
| RIKEN BDR-Otsuka Pharmaceutical Collaboration | | Igor Kondrychyn |
| Cancerfonden | 2018/1154 | Christer Betsholtz |
| Knut and Alice Wallenberg Foundation | 2018.0218 | Christer Betsholtz |
| Swedish Brain Foundation | ALZ2022-0005 | Christer Betsholtz |
| Leducq Foundation | 23CVD02 | Christer Betsholtz |
| Japan Society for the Promotion of Science | 22H05168 | Li-Kun Phng |

The funders had no role in study design, data collection and interpretation, or the decision to submit the work for publication.

## Author contributions

Igor Kondrychyn, Data curation, Formal analysis, Investigation, Methodology, Writing – original draft; Liqun He, Formal analysis, Methodology; Haymar Wint, Investigation; Christer Betsholtz, Supervision, Funding acquisition; Li-Kun Phng, Conceptualization, Resources, Formal analysis, Supervision, Funding acquisition, Investigation, Visualization, Writing – original draft, Project administration, Writing – review and editing

### Author ORCIDs

Igor Kondrychyn  https://orcid.org/0000-0002-2268-7291
Christer Betsholtz  https://orcid.org/0000-0002-8494-971X
Li-Kun Phng  https://orcid.org/0000-0001-8523-9958

### Ethics

All animal experiments were approved by the Institutional Animal Care and Use Committee at RIKEN Kobe Branch (IACUC).

Reviewer #1 (Public review): https://doi.org/10.7554/eLife.98612.3.sa1
Reviewer #3 (Public review): https://doi.org/10.7554/eLife.98612.3.sa2
Author response https://doi.org/10.7554/eLife.98612.3.sa3

# Additional files

### Supplementary files

Supplementary file 1. Plasmids used in this study.

Supplementary file 2. Oligonucleotides used in this study.

MDAR checklist

### Data availability

All data generated or analyzed in this study are included in the manuscript and associated source data files. The scRNA-seq raw data from this study is available at NCBI's Gene Expression Omnibus database (accession number GSE262232). Raw image files used to generate figures can be accessed in figshare.

The following datasets were generated:

| Author(s) | Year | Dataset title | Dataset URL | Database and Identifier |
|---|---|---|---|---|
| Phng LK | 2024 | Single cell RNA-seq of zebrafish endothelial cells | https://www.ncbi.nlm.nih.gov/geo/query/acc.cgi?acc=GSE262232 | NCBI Gene Expression Omnibus, GSE262232 |
| Phng LK, Kondrychyn I, He L, Betsholtz C, Wint H | 2025 | Combined forces of hydrostatic pressure and actin polymerization drive endothelial tip cell migration and sprouting angiogenesis. | https://doi.org/10.6084/m9.figshare.28441904 | figshare, 10.6084/m9.figshare.28441904 |

The following previously published dataset was used:

| Author(s) | Year | Dataset title | Dataset URL | Database and Identifier |
|---|---|---|---|---|
| Gurung S, Restrepo NK, Chestnut BK, Klimkaite L, Sumanas S | 2022 | Single-cell transcriptomic analysis of vascular endothelial cells in zebrafish embryos | https://www.ncbi.nlm.nih.gov/geo/query/acc.cgi?acc=GSE202912 | NCBI Gene Expression Omnibus, GSE202912 |

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
