## [Editor Report · eLife Assessment]

This study **convincingly** shows that aquaporin-mediated cell migration plays a key role in blood vessel formation during zebrafish development. In particular, the article implicates hydrostatic pressure and water flow as mechanisms controlling endothelial cell migration during angiogenic sprouting. This **fundamental** study is highly novel and significantly advances our understanding of cell migration during morphogenesis. As such, this work will be of great interest to developmental and cell biologists working on organogenesis, angiogenesis, and cell migration.

---

## [Referee Report · Reviewer #1 (Public review)]

Summary:

The paper details a study of endothelial cell vessel formation during zebrafish development. The results focus on the role of aquaporins, which mediate the flow of water across the cell membrane, leading to cell movement. The authors show that actin and water flow together drive endothelial cell migration and vessel formation. If any of these two elements are perturbed, there are observed defects in vessels. Overall, the paper significantly improves our understanding of cell migration during morphogenesis in organisms.

Strengths:

The data are extensive and are of high quality. There is a good amount of quantification with convincing statistical significance. The overall conclusion is justified given the evidence.

Weaknesses:

There are two weaknesses, which if addressed, would improve the paper.

(1) The paper focuses on aquaporins, which while mediates water flow, cannot drive directional water flow. If the osmotic engine model is correct, then ion channels such as NHE1 are the driving force for water flow. Indeed this water is shown in previous studies. Moreover, NHE1 can drive water intake because the export of H+ leads to increased HCO3 due to reaction between CO2+H2O, which increases the cytoplasmic osmolarity (see Li, Zhou and Sun, Frontiers in Cell Dev. Bio. 2021). If NHE cannot be easily perturbed in zebrafish, it might be of interest to perturb Cl channels such as SWELL1, which was recently shown to work together with NHE (see Zhang, et al, Nat. Comm. 2022).

After revision, this concern has been addressed.

(2) In some places the discussion seems a little confusing where the text goes from hydrostatic pressure to osmotic gradient. It might improve the paper if some background is given. For example, mention water flow follows osmotic gradients, which will build up hydrostatic pressure. The osmotic gradients across the membrane are generated by active ion exchangers. This point is often confused in literature and somewhere in the intro, this could be made clearer.

After revision, this concern has been addressed.

---

## [Referee Report · Reviewer #3 (Public review)]

Summary:

Kondrychyn and colleagues describe the contribution of two Aquaporins Aqp1a.1 and Aqp8a.1 towards angiogenic sprouting in the zebrafish embryo. By whole-mount in situ hybridization, RNAscope and scRNA-seq, they show that both genes are expressed in endothelial cells in partly overlapping spatiotemporal patterns. Pharmacological inhibition experiments indicate a requirement for VEGR2 signaling (but not Notch) in transcriptional activation.

To assess the role of both genes during vascular development the authors generate genetic mutations. While homozygous single mutants appear normal, aqp1a.1;aqp8a.1 double mutants exhibit defects in EC sprouting and ISV formation.

At the cellular level, the aquaporin mutants display a reduction of filopodia in number and length. Furthermore, a reduction in cell volume is observed indicating a defect in water uptake.

The authors conclude, that polarized water uptake mediated by aquaporins is required for the initiation of endothelial sprouting and (tip) cell migration during ISV formation. They further propose that water influx increases hydrostatic pressure within the cells which may facilitate actin polymerization and formation membrane protrusions.

In the revised version of the manuscript the authors have added data which show that inhibition of swell-induced chloride channels mimics aqp mutant phenotypes, giving credence to the model that water influx via aquaporins is driven by an osmotic gradient.

Strengths:

The authors provide a detailed analysis of Aqp1a.1 and Aqp8a.1 during blood vessel formation in vivo, using zebrafish intersomitic vessels as a model. State-of-the-art imaging demonstrates an essential role aquaporins in different aspects of endothelial cell activation and migration during angiogenesis.

Weaknesses:

With respect to the connection between Aqp1/8 and actin polymerization/filopodia formation, the evidence appears preliminary and the authors' interpretation is guided by evidence from other experimental systems.

After revision, the authors have addressed all other concerns

---

## [Author Response]

The following is the authors’ response to the original reviews.

**Reviewer #1 (Public Review):**
Summary:This paper details a study of endothelial cell vessel formation during zebrafish development. The results focus on the role of aquaporins, which mediate the flow of water across the cell membrane, leading to cell movement. The authors show that actin and water flow together drive endothelial cell migration and vessel formation. If any of these two elements are perturbed, there are observed defects in vessels. Overall, the paper significantly improves our understanding of cell migration during morphogenesis in organisms.Strengths:The data are extensive and are of high quality. There is a good amount of quantification with convincing statistical significance. The overall conclusion is justified given the evidence.Weaknesses:There are two weaknesses, which if addressed, would improve the paper.(1) The paper focuses on aquaporins, which while mediates water flow, cannot drive directional water flow. If the osmotic engine model is correct, then ion channels such as NHE1 are the driving force for water flow. Indeed this water is shown in previous studies. Moreover, NHE1 can drive water intake because the export of H+ leads to increased HCO3 due to the reaction between CO2+H2O, which increases the cytoplasmic osmolarity (see Li, Zhou and Sun, Frontiers in Cell Dev. Bio. 2021). If NHE cannot be easily perturbed in zebrafish, it might be of interest to perturb Cl channels such as SWELL1, which was recently shown to work together with NHE (see Zhang, et al, Nat. Comm. 2022).(2) In some places the discussion seems a little confusing where the text goes from hydrostatic pressure to osmotic gradient. It might improve the paper if some background is given. For example, mention water flow follows osmotic gradients, which will build up hydrostatic pressure. The osmotic gradients across the membrane are generated by active ion exchangers. This point is often confused in literature and somewhere in the intro, this could be made clearer.
**Reviewer #1 (Recommendations For The Authors):**
(1) The paper focuses on aquaporins, which while mediating water flow, cannot drive directional water flow. If the osmotic engine model is correct, then ion channels such as NHE1 are the driving force for water flow. Indeed this water is shown in previous studies. Moreover, NHE1 can drive water intake because the export of H+ leads to increased HCO3 due to the reaction between CO2+H2O, which increases the cytoplasmic osmolarity (see Li, Zhou and Sun, Frontiers in Cell Dev. Bio. 2021). If NHE cannot be easily perturbed in zebrafish, it might be of interest to perturb Cl channels such as SWELL1, which was recently shown to work together with NHE (see Zhang, et al, Nat. Comm. 2022).

We thank Reviewer #1 for this very important comment and the suggestion to examine the function of ion channels in establishing an osmotic gradient to drive directional flow. We have taken on board the reviewer’s suggestion and examined the expression of NHE1 and SWELL1 in endothelial cells using published scRNAseq of 24 hpf ECs (Gurung et al, 2022, Sci. Rep.). We found that *slc9a1a, slc9a6a, slc9a7, slc9a8, lrrc8aa* and *lrrc8ab* are expressed in different endothelial subtypes. To examine the function of NHE1 and SWELL1 in endothelial cell migration, we used the pharmacological compounds, 5-(N-ethyl-Nisopropyl)amiloride (EIPA) and DCPIB, respectively. While we were unable to observe an ISV phenotype after EIPA treatment at 5, 10 and 50µM, we were able to observe impaired ISV formation after DCPIB treatment that was very similar to that observed in Aquaporin mutants. We were very encouraged by these results and proceeded to perform more detailed experiments whose results have yielded a new figure (Figure 6) and are described and discussed in lines 266 to 289 and 396 to 407, respectively, in the revised manuscript.

(2) In some places the discussion seems a little confusing where the text goes from hydrostatic pressure to osmotic gradient. It might improve the paper if some background is given. For example, mention water flow follows osmotic gradients, which will build up hydrostatic pressure. The osmotic gradients across the membrane are generated by active ion exchangers. This point is often confused in literature and somewhere in the intro, this could be made clearer.

Thank you for pointing out the deficiency in explaining how osmotic gradients drive water flow to build up hydrostatic pressure. We have clarified this in lines 50, 53 - 54 and 385.

The two recommendations listed above would improve the paper. They are however not mandatory. The paper would be acceptable with some clarifying rewrites. I am not an expert on zebrafish genetics, so it might be difficult to perturb ion channels in this model organism. Have the authors tried to perturb ion channels in these cells?

We hope that our attempts at addressing Reviewer’s 1 comments are satisfactory and sufficient to clarify the concerns outlined.

**Reviewer #2 (Public Review):**
Summary:Directional migration is an integral aspect of sprouting angiogenesis and requires a cell to change its shape and sense a chemotactic or growth factor stimulus. Kondrychyn I. et al. provide data that indicate a requirement for zebrafish aquaporins 1 and 8, in cellular water inflow and sprouting angiogenesis. Zebrafish mutants lacking aqp1a.1 and aqp8a.1 have significantly lower tip cell volume and migration velocity, which delays vascular development. Inhibition of actin formation and filopodia dynamics further aggravates this phenotype. The link between water inflow, hydrostatic pressure, and actin dynamics driving endothelial cell sprouting and migration during angiogenesis is highly novel.Strengths:The zebrafish genetics, microscopy imaging, and measurements performed are of very high quality. The study data and interpretations are very well-presented in this manuscript.Weaknesses:Some of the mechanobiology findings and interpretations could be strengthened by more advanced measurements and experimental manipulations. Also, a better comparison and integration of the authors' findings, with other previously published findings in mice and zebrafish would strengthen the paper.

We thank Reviewer #2 for the critique that the paper can be strengthened by more advanced measurements and experimental manipulations. One of the technical challenges that we face is how to visualize and measure water flow directly in the zebrafish. We have therefore taken indirect approaches to assess water abundance in endothelial cells in vivo. One approach was to measure the diffusion of GEM nanoparticles in tip cell cytoplasm in wildtype and Aquaporin mutants, but results were inconclusive. The second was to measure the volume of tip cells, which should reflect water in/outflow. As the second approach produced clear and robust differences between wildtype ECs, ECs lacking Aqp1a.1 and Aqp8a.1 and ECs overexpressing Aqp1a.1 (revised Fig. 5), we decided to present these data in this manuscript.

We have also taken Reviewer 2 advice to better incorporate previously published data in our discussion (see below and lines 374 to 383 of the revised manuscript).

**Reviewer #2 (Recommendations For The Authors):**
I have a few comments that the authors may address to further improve their manuscript analysis, quality, and impact.Major comments:(1) Citation and discussion of published literature

The authors have failed to cite and discuss recently published results on the role of aqp1a.1 and aqp8a.1 in ISV formation and caliber in zebrafish (Chen C et al. Cardiovascular Research 2024). That study showed a similar impairment of ISV formation when aqp1a.1 is absent but demonstrated a stronger phenotype on ISV morphology in the absence of aqp8a.1 than the current manuscript by Kondrychyn I et al. Furthermore, Chen C et al show an overall decrease in ISV diameter in single aquaporin mutants suggesting that the cell volume of all ECs in an ISV is affected equally. Given this published data, are ISV diameters affected in single and double mutants in the current study by Kondrochyn I et al? An overall effect on ISVs would suggest that aquaporin-mediated cell volume changes are not an inherent feature of endothelial tip cells. The authors need to analyse/compare and discuss all differences and similarities of their findings to what has been published recently.

We apologise for having failed and discussed the recently published paper by Chen et al. This has been corrected and discussed in lines 374 to 383.

In the paper by Chen et al, the authors describe a role of Aqp1a.1 and Aqp8a.1 in regulating ISV diameter (ISV diameter was analysed at 48 hpf) but they did not examine the earlier stages of sprouting angiogenesis between 20 to 30 hpf, which is the focus of our study. We therefore cannot directly compare the ISV phenotypes with theirs. Nevertheless, we recognise that there are differences in ISV phenotypes from 2 dpf. For example, they did not observe incompletely formed or missing ISVs at 2 and 3 dpf, which we clearly observe in our study. This could be explained by differences in the mutations generated. In Chen et al., the sgRNA used targeted the end of exon 2 that resulted in the generation of a 169 amino acid truncated aqp1a.1 protein. However, in our approach, our sgRNA targeted exon 1 of the gene that resulted in a truncated aqp1a.1 protein that is 76 amino acid long. As for the *aqp8a.1* zebrafish mutant that we generated, our sgRNA targeted exon 1 of the gene that resulted in a truncated protein that is 73 amino acids long. In Chen et al., the authors did not generate an *aqp8a.1* mutant but instead used a crispant approach, which leads to genetic mosaicism and high experimental variability.

Following the reviewer’s suggestion, we have now measured the diameters of arterial ISVs (aISVs) and venous ISVs (vISVs) in *aqp1a.1-/-*, *aqp8a.1-/-* and *aqp1a.1-/-*;*aqp8a.1-/-* zebrafish. In our lab, we always make a distinction between aISVs and vISVs are their diameters are significantly different from each other. The results are in Fig S11A. While we corroborate a decrease in diameter in both aISVs and vISVs in single *aqp1a.1-/-* and double *aqp1a.1-/-*;*aqp8a.1-/-*.zebrafish, we observed a slight increase in diameter in both aISVs and vISVs in *aqp8a.1-/-* zebrafish at 2 dpf. We also measured the diameter of aISV and vISV in *Tg(fli1ep:aqp1a.1-mEmerald)* and *Tg(fli1ep:aqp8a.1-mEmerald)* zebrafish at 2 dpf (Fig S11B) and unlike in Chen et al., we could not detect a difference in the diameter between control and aqp1a.1- or aqp8a.1-overexpressing endothelial cells.

We also would also like to point out that, because ISVs are incompletely formed or are missing in *aqp1a.1-/-*;*aqp8a.1-/-* zebrafish (Fig. 3G – L), blood flow is most likely altered in the zebrafish trunk of these mutants, and this can have a secondary effect on blood vessel calibre or diameter. In fact, we often observed wider ISVs adjacent to unperfused ISVs (Fig. 3J) as more blood flow enters the lumenized ISV. Therefore, to determine the cell autonomous function of Aquaporin in mediating cell volume changes in vessel diameter regulation, one would need to perform cell transplantation experiments where we would measure the volume of single *aqp1a.1-/-*;*aqp8a.1-/-* endothelial cells in wildtype embryos with normal blood flow. As this is beyond the scope of the present study, we have not done this experiment during the revision process.

(2) Expression of aqp1a.1 and aqp8a.1The quantification shown in Figure 1G shows a relative abundance of expression between tip and stalk cells. However, it seems aqp8a.1 is almost never detected in most tip cells. The authors could show in addition, the % of Tip and stalk cells with detectable expression of the 2 aquaporins. It seems aqp8a1 is really weakly or not expressed in the initial stages. Ofcourse the protein may have a different dynamic from the RNA.

We would like to clarify that *aqp8a.1* mRNA is not detected in tip cells of newly formed ISVs at 20hpf. At 22 hpf, it is expressed in both tip cells (22 out of 23 tip cells analysed) and stalk cells of ISVs at 22hpf. This is clarified in lines 107 - 109. We also include below a graph showing that although *aqp8a.1* mRNA is expressed in tip cells, its expression is higher in stalk cells.

Could the authors show endogenously expressed or tagged protein by antibody staining? The analysis of the Tg(fli1ep:aqp8a.1-mEmerald)rk31 zebrafish line is a good complement, but unfortunately, it does not reveal the localization of the endogenously expressed protein. Do the authors have any data supporting that the endogenously expressed aqp8a.1 protein is present in sprouting tip cells?

We tested several antibodies against AQP1 (Alpha Diagnostic International, AQP11-A; ThermoFisher Scientific, MA1-20214; Alomone Labs, AQP-001) and AQP8 (Sigma Aldrich, SAB 1403559; Alpha Diagnostic International, AQP81-A; Almone Labs, AQP-008) but unfortunately none worked. As such, we do not have data demonstrating endogenous expression and localisation of Aqp1a.1 and Aqp8a.1 proteins in endothelial cells.

Could the authors perform F0 CRISPR/Cas9 mediated knockin of a small tag (i.e. HA epitope) in zebrafish and read the endogenous protein localization with anti-HA Ab?

CRISPR/Cas9 mediated in-frame knock-in of a tag into a genomic locus is a technical challenge that our lab has not established. We therefore cannot do this experiment within the revision period.

Given the double mutant phenotypic data shown, is aqp8a.1 expression upregulated and perhaps more important in aqp1a.1 mutants?

In our analysis of *aqp1a.1* homozygous zebrafish, there is a slight _down_regulation in *aqp8a.1* expression (Fig. S5C). Because the loss of Aqp1a.1 leads to a stronger impairment in ISV formation than the loss of Aqp8a.1 (see Fig. S6F, G, I and J), we believe that Aqp1a.1 has a stronger function than Aqp8a.1 in EC migration during sprouting angiogenesis.

Regarding the regulation of expression by the Vegfr inhibitor Ki8751, does this inhibitor affect Vegfr/ERK signalling in zebrafish and the sprouting of ISVs significantly?

ki8751 has been demonstrated to inhibit ERK signalling in tip cells in the zebrafish by Costa et al., 2016 in Nature Cell Biology. In our experiments, treatment with 5 µM ki8751 for 6 hours from 20 hpf also inhibited sprouting of ISVs.

The data presented suggest that tip cells overexpressing aqp1a.1-mEmerald (Figure 2C) need more than 6 times longer to migrate the same distance as tip cells expressing aqp8a.1mEmerald (Figure 2D). How does this compare with cells expressing only Emerald? A similar time difference can be seen in Movie S1 and Movie S2. Is it just a coincidence? Could aqp8a.1, when expressed at similar levels than aqp1a, be more functional and induce faster cell migration? These experiments were interpreted only for the localization of the proteins, but not for the potential role of the overexpressed proteins on function. Chen C et al. Cardiovascular Research 2024 also has some Aqp overexpression data.

The still images prepared for Fig. 2 C and D were selected to illustrate the localization of Aqp1a.1-mEmerald and Aqp8a.1-mEmerald at the leading edge of migrating tip cells. We did not notice that the tip cell overexpressing Aqp1a.1-mEmerald (Figure 2C) needed more than 6 times longer to migrate the same distance as the tip cell expressing aqp8a.1-mEmerald (Figure 2D), which the reviewer astutely detected. To ascertain whether there is a difference in migration speed between Aqp1a.1-mEmerald and Aqp8a.1-mEmerald overexpressing endothelial cells, we measured tip cell migration velocity of three ISVs from *Tg(fli1ep:aqp1a.1-mEmerald)* and *Tg(fli1ep:aqp8a.1-mEmerald)* zebrafish during the period of ISV formation (24 to 29 hpf) using the Manual Tracking plugin in Fiji. As shown in the graph, there is no significant difference in the migration speed of ECs overexpressing Aqp1a.1-mEmerald and Aqp8a.1-mEmerald, suggesting that Aqp8a.1-overexpressing cells migrate at a similar rate as Aqp1a.1-overexpressing cells. As we have not generated a *Tg(fli1ep:mEmerald)* zebrafish line, we are unable to determine whether endothelial cells migrate faster in *Tg(fli1ep:aqp1a.1mEmerald)* and *Tg(fli1ep:aqp8a.1-mEmerald)* zebrafish compared to endothelial cell expressing only mEmerald. As for the observation that tip cells overexpressing aqp1a.1mEmerald (Figure 2C) need more than 6 times longer to migrate the same distance as tip cells expressing aqp8a.1-mEmerald, we can only surmise that it is coincidental that the images selected “showed” faster migration of one ISV from *Tg(fli1ep:aqp8a.1-mEmerald)* zebrafish. We do not know whether the Aqp1a.1 and Aqp8a.1 are overexpressed to the same levels in *Tg(fli1ep:aqp1a.1mEmerald)* and *Tg(fli1ep:aqp8a.1-mEmerald)* zebrafish.

We would also like to point out that when we analysed the lengths of ISVs at 28 hpf in *aqp1a.1-/-* and *aqp8a.1-/-* zebrafish, ISVs were shorter in *aqp1a.1-/-* zebrafish compared to *aqp8a.1-/-* zebrafish (Fig. S6 F to J). These results indicate that the loss of Aqp1a.1 function causes slower migration than the loss of aqp8a.1 function, and suggest that Aqp1a.1 induces faster endothelial cell migration that Aqp8a.1.

**Author response image 2. sa3fig2:** 

The data on Aqps expression after the Notch inhibitor DBZ seems unnecessary, and is at the moment not properly discussed. It is also against what is set in the field. aqp8a.1 levels seem to increase only 24h after DBZ, not at 6h, and still authors conclude that Notch activation inhibits aqp8a.1 expression (Line 138-139). In the field, Notch is considered to be more active in stalk cells, where aqp8a.1 expression seems higher (not lower). Maybe the analysis of tip vs stalk cell markers in the scRNAseq data, and their correlation with Hes1/Hey1/Hey2 and aqp1 vs aqp8 mRNA levels will be more clear than just showing qRT-PCR data after DBZ.

As our scRNAseq data did not include ECs from earlier during development when ISVs are developing, we have analysed of scRNAseq data of 24 hpf endothelial cells published by Gurung et al, 2022 in Scientific Reports during the revision of this manuscript. However, we are unable to detect separate clusters of tip and stalk cells. As such, we are unable to correlate *hes1/hey1/hey2* expression (which would be higher in stalk cells) with that of *aqp1a.1/aqp8a.1*. Also, we have decided to remove the DBZ-treatment results from our manuscript as we agree with the two reviewers that they are unnecessary.

The paper would also benefit from some more analysis and interpretation of available scRNAseq data in development/injury/disease/angiogenesis models (zebrafish, mice or humans) for the aquaporin genes characterized here. To potentially raise a broader interest at the start of the paper.

We thank the reviewer for suggesting examining aquaporin genes in other angiogenesis/disease/regeneration models to expand the scope of aquaporin function. We will do this in future studies.

(3) Role of aqp1a.1 and aqp8a.1 on cytoplasmic volume changes and related phenotypesIn Figure 5 the authors show that Aqp1/Aqp8 mutant endothelial tip cells have a lower cytoplasmic volume than tip cells from wildtype fish. If aquaporin-mediated water inflow occurs locally at the leading edge of endothelial tip cells (Figure 2, line 314-318), why doesn't cytoplasmic volume expand specifically only at that location (as shown in immune cells by Boer et al. 2023)? Can the observed reduction in cytoplasmic volume simply be a side-effect of impaired filopodia formation (Figure 4F-I)?

We believe that water influx not only expands filopodia but also the leading front of tip cells (see bracket region in Fig. 4D), where Aqp1a.1-mEmerald/Aqp8a.1-mEmerald accumulate (Fig. 2), to generate an elongated protrusion and forward expansion of the tip cell. The decrease in cytoplasmic volume observed in the *aqp1a.1;aqp8a.1* double mutant zebrafish is a result of decreased formation of these elongated protrusions at the leading front of migration tip cells as shown in Fig. 4E (compare to Fig. 4D), not from just a decrease in filopodia number. In fact, in the method used to quantify cell volume, mEmerald/EGFP localization is limited to the cytoplasm and does not label filopodia well (compare mEmerald/EGFP in green with membrane tagged-mCherry in Fig. 5A - C). The volume measured therefore reflects cytoplasmic volume of the tip cell, not filopodia volume.

Do the authors have data on cytoplasmic volume changes of endothelial tip cells in latrunculin B treated fish? The images in Figures 6 A,B suggest that there is a difference in cell volume upon lat b treatment only.

No, unfortunately we have not performed single cell labelling and measurement of tip cells in Latrunculin B-treated embryos. We can speculate that as there is a decrease in actindriven membrane protrusions in this experiment, one would also expect a decrease in cell volume as the reviewer has observed.

(4) Combined loss of aquaporins and actin-based force generation.Lines 331-332 " we show that hydrostatic pressure is the driving force for EC migration in the absence of actin-based force generation"....better leave it more open and stick to the data. The authors show that aquaporin-mediated water inflow partially compensates for the loss of actin-based force generation in cell migration. Not that it is the key driving/rescuing force in the absence of actin-based force.

We have changed it to “we show that hydrostatic pressure *can generate force* for EC migration in the absence of actin-based force generation” in line 348.

(5) Aquaporins and their role in EC proliferationIn the study by Phnk LK et al. 2013, the authors have shown that proliferation is not affected when actin polymerization or filopodia formation is inhibited. However, in the current manuscript by Kondrychyn I. et al. this has not been analysed carefully. In Movie S4 the authors indicate by arrows tip cells that fail to invade the zebrafish trunk demonstrating a severe defect of sprouting initiation in these mutants. Yet, when only looking at ISVs that reach the dorsal side in Movie S4, it appears that they are comprised of fewer EC nuclei/ISV than the ISVs in Movie S3. At the beginning of DLAV formation, most ISVs in control Movie S3 consist of 3-4 EC nuclei, while in double mutants Movie S4 it appears to be only 2-3 EC nuclei. At the end of the Movie S4, one ISV on the left side even appears to consist of only a single EC when touching the dorsal roof. The authors provide convincing data on how the absence of aquaporin channels affects sprouting initiation and migration speed, resulting in severe delay in ISV formation. However, the authors should also analyse EC proliferation, as it may also be affected in these mutants, and may also contribute to the observed phenotype. We know that effects on cell migration may indirectly change the number of cells and proliferation at the ISVs, but this has not been carefully analysed in this paper.

We thank the reviewer for highlighting the lack of information on EC number and division in the aquaporin mutants. We have now quantified EC number in ISVs that are fully formed (i.e. connecting the DA or PCV to the DLAV) at 2 and 3 dpf and the results are displayed in Figure S10A and B. At 2 dpf, there is a slight but significant reduction in EC number in both aISVs and vISVs in *aqp1a.1-/-* zebrafish and an even greater reduction in the double aqp1a. *aqp1a.1/-*;*aqp8a.1-/-* zebrafish. No significant change in EC number was observed in *aqp8a.1-/-* zebrafish. EC number was also significantly decreased at 3 dpf for *aqp1a.1-/-*, *aqp8a.1-/-* and *aqp1a.1-/-*;*aqp8a.1-/-* zebrafish. The decreased in EC number per ISV may therefore contribute to the observed phenotype.

We have also quantified the number of cell divisions during sprouting angiogenesis (from 21 to 30 hpf) to assess whether the lack of Aquaporin function affects EC proliferation. This analysis shows that there is no significant difference in the number of mitotic events between *aqp1a.1+/-*; *aqp8a.1+/-* and *aqp1a.1-/-*;*aqp8a.1-/-* zebrafish (Figure S10 C), suggesting that the reduction in EC number is not caused by a decrease in EC proliferation.

These new data are reported on lines 198 to 205 of the manuscript.

Minor comments:- Figure 3K data seems not to be necessary and even partially misleading after seeing Figure 3E. Fig. 3E represents the true strength of the phenotype in the different mutants.

Figure 3K has been removed from Figure 3.

- Typo Figure 3L (VII should be VI).

Thank you for spotting this typo. VII has been changed to VI.

- Line 242: The word "required" is too strong because there is vessel formation without Aqps in endothelial cells.

This has been changed to “ …Aqp1a.1 and Aqp8a.1 regulate sprouting angiogenesis…” (lines 238 - 239).

- From Figure S2, the doublets cluster should be removed.

We have performed a new analysis of 24 hpf, 34hpf and 3 dpf endothelial cells scRNAseq data (the previous analysis did not consist of 24 hpf endothelial cells). The doublets cluster is not included in the UMAP analysis.

- Better indicate the fluorescence markers/alleles/transgenes used for imaging in Figures 6A-D.

The transgenic lines used for this experiment are now indicated in the figure (this figure is now Figure 7).

**Reviewer #3 (Public Review):**
Summary:Kondrychyn and colleagues describe the contribution of two Aquaporins Aqp1a.1 and Aqp8a.1 towards angiogenic sprouting in the zebrafish embryo. By whole-mount in situ hybridization, RNAscope, and scRNA-seq, they show that both genes are expressed in endothelial cells in partly overlapping spatiotemporal patterns. Pharmacological inhibition experiments indicate a requirement for VEGR2 signaling (but not Notch) in transcriptional activation.To assess the role of both genes during vascular development the authors generate genetic mutations. While homozygous single mutants appear normal, aqp1a.1;aqp8a.1 double mutants exhibit defects in EC sprouting and ISV formation.

At the cellular level, the aquaporin mutants display a reduction of filopodia in number and length. Furthermore, a reduction in cell volume is observed indicating a defect in water uptake.

The authors conclude, that polarized water uptake mediated by aquaporins is required for the initiation of endothelial sprouting and (tip) cell migration during ISV formation. They further propose that water influx increases hydrostatic pressure within the cells which may facilitate actin polymerization and formation membrane protrusions.

Strengths:

The authors provide a detailed analysis of Aqp1a.1 and Aqp8a.1 during blood vessel formation in vivo, using zebrafish intersomitic vessels as a model. State-of-the-art imaging demonstrates an essential role in aquaporins in different aspects of endothelial cell activation and migration during angiogenesis.

Weaknesses:

With respect to the connection between Aqp1/8 and actin polymerization/filopodia formation, the evidence appears preliminary and the authors' interpretation is guided by evidence from other experimental systems.

**Reviewer #3 (Recommendations For The Authors):**
Figure 1 H, J:The differential response of aqp1/-8 to ki8751 vs DBZ after 6h treatment is quite obvious. Why do the authors show the effect after 24h? The effect is more likely than not indirect.

We agree with the reviewer and we have now removed 24 hour Ki8751 treatment and all DBZ treatments from Figure 1.

Figure 2:According to the authors' model anterior localization of Aqp1 protein is critical. The authors perform transient injections to mosaically express Aqp fusion proteins using an endothelial (fli1) promoter. For the interpretation, it would be helpful to also show the mCherry-CAAX channel in separate panels. From the images, it is not possible to discern how many cells we are looking at. In particular the movie in panel D may show two cells at the tip of the sprout. A marker labelling cell-cell junctions would help. Furthermore, the authors are using a strong exogenous promoter, thus potentially overexpressing the fusion protein, which may lead to mislocalization. For Aqp1a.1 an antibody has been published to work in zebrafish (e.g. Kwong et al., Plos1, 2013).

We would like to clarify that we generated transgenic lines - *Tg(fli1ep:aqp1a.1-mEmerald)* and *Tg(fli1ep:aqp8a.1-mEmerald)* - to visualize the localization of Aqp1a.1 and Aqp8a.1 in endothelial cells, and the images displayed in Fig. 2 are from the transgenic lines (not transient, mosaic expression).

To aid visualization and interpretation, we have now added mCherry-CAAX only channel to accompany the Aqp1a.1/Aqp8a.1-mEmerald channel in Fig. 2A and B. To discern how many cells there are in the ISVs at this stage, we have crossed *Tg(fli1ep:aqp1a.1-mEmerald)* and *Tg(fli1ep:aqp8a.1-mEmerald)* zebrafish to *TgKI(tjp1a-tdTomato)pd1224* (Levic et al., 2021) to visualize ZO1 at cell-cell junction. However, because tjp1-tdTomato is expressed in all cell types including the skin that lies just above the ISV and the signal in ECs in ISVs is very weak at 22 to 25 hpf, it was very difficult to obtain good quality images that can properly delineate cell boundaries to determine the number of cells in the ISVs at this early stage. Instead, we have annotated endothelial cell boundaries based on more intense mCherryCAAX fluorescence at cell-cell borders, and from the mosaic expression of mCherryCAAX that is intrinsic to the *Tg(kdrl:ras-mCherry)s916* zebrafish line.

In Fig. 2D, there are two endothelial cells in the ISV during the period shown but there is only 1 cell occupying the tip cell position i.e. there is one tip cell in this ISV. Unlike the mouse retina where it has been demonstrated that two endothelial cells can occupy the tip cell position side-by-side (Pelton et al., 2014), this is usually not observed in zebrafish ISVs. This is demonstrated in Movie S3, where it is clear that one nucleus (belonging to the tip cell) occupies the tip of the growing ISV. The accumulation of intracellular membranes is often observed in tip cells that may serve as a reservoir of membranes for the generation of membrane protrusions at the leading edge of tip cells.

We agree that by generating transgenic *Tg(fli1ep:aqp1a.1-mEmerald)* and *Tg(fli1ep:aqp8a.1mEmerald)* zebrafish, Aqp1a.1 and Aqp8a.1 are overexpressed that may affect their localization. The eel anti-Aqp1a.1 antibody used in (Kwong et la., 2013) was a gift from Dr. Gordon Cramb, Univ. of St Andrews, Scotland and it was first published in 2001. This antibody is not available commercially. Instead, we have tried to several other antibodies against AQP1 (Alpha Diagnostic International , AQP11-A; ThermoFisher Scientific, MA120214; Alomone Labs, AQP-001) and AQP8 (Sigma Aldrich, SAB 1403559; Alpha Diagnostic International, AQP81-A; Almone Labs, AQP-008) but unfortunately none worked. As such, we cannot compare localization of Aqp1a.1-mEmerald and Aqp8a.1-mEmerald with the endogenous proteins.

Figure 3:E: the quantification is difficult to read. Wouldn't it be better to set the y-axis in % of the DV axis? (see also Figure S6).

We would like to show the absolute length of the ISVs, and to illustrate that the ISV length decreases from anterior to posterior of the zebrafish trunk. We have increased the size of Fig. 3E to enable easier reading of the bars.

K: This quantification appears arbitrary.

We have removed this panel from Figure 3.

G-J: The magenta channel is difficult to see. Is the lifeact-mCherry mosaic? In panel J there appears to be a nucleus between the sprout and the DLAV. It would be helpful to crop the contralateral side of the image.

No, the *Tg(fli1:Lifeact-mCherry)* line is not mosaic. The “missing” vessels are not because of mosaicism in transgene but because of truncated ISVs that is a phenotype of loss Aquaporin function. We have changed the magenta channel to grey and hope that by doing so, the reviewer will be able to see the shape of the blood vessels more clearly. We would like to leave the contralateral side in the images, as it shows that the defective vessel is only on one side of body. Furthermore, when we tried to remove it (reducing the number of Z-stacks) neighbour ISV looks incomplete because the embryos were not mounted flat. To clarify what the nucleus between the sprout and the DLAV is, we have indicated that it is that of the contralateral ISV.

L: I do not quite understand the significance of the different classes of phenotypes. Do the authors propose different morphogenetic events or contexts of how these differences come about?

Here, we report the different types of ISV phenotypes that we observe in 3 dpf *aqp1a.1-/-*; *aqp8a.1-/-* zebrafish (Fig. 3 and Fig. S7). As demonstrated in Fig. 4, most of the phenotypes can be explained by the delayed emergence of tip cells from the dorsal aorta and slower tip cell migration. However, in some instances, we also observed retraction of tip cells (Movie S4) and failure of tip cells to emerge from the dorsal aorta or endothelial cell death (see attached figure on page 14), which can give rise to the Class II phenotype. In the dominant class I phenotype (in contrast to class II), secondary sprouting from the posterior cardinal vein is unaffected, and the secondary sprout migrates dorsally passing the level of horizontal myoseptum but cannot complete the formation of vISV (it stops beneath the spinal cord). The Class III phenotype appears to result from a failure of the secondary sprout to fuse with the regressed primary ISV. In the Class IV phenotype, the ventral EC does not maintain a connection to the dorsal aorta. We did not examine how Class III and IV phenotypes arise in detail in this current study.

**Author response image 3. sa3fig3:** 

Figure 4:This figure nicely demonstrates the defects in cell behavior in aqp mutants.In panel F it would be helpful to show the single channels as well as the merge.

We have now added single channels for PLCd1PH and Lifeact signal in panels F and G.

In Figure 1 the authors argue that the reduction of Aqp1/8 by VEGFR2 inhibition may account for part of that phenotype. In turn, the aqp phenotype seems to resemble incomplete VEGFR2 inhibition. The authors should check whether expression Aqp1Emerald can partially rescue ki8751 inhibition.

To address the reviewer’s comment, we have treated *Tg(fli1ep:Aqp1-Emerald)* embryos with ki8751 from 20 hpf for 6 hours but we were unable to observe a rescue in sprouting. It could be because VEGFR2 inhibition also affects other downstream signalling pathways that also control cell migration as well as proliferation.

Based on previous studies (Loitto et al.; Papadopoulus et al.) the authors propose that also in ISVs aquaporin-mediated water influx may promote actin polymerization and thereby filopodia formation. However, while the effect on filopodia number and length is well demonstrated, the underlying cause is less clear. For example, filopodia formation could be affected by reduced cell polarization. This can be tested by using a transgenic golgi marker (Kwon et al., 2016).

We have examined tip cell polarity of wildtype, *aqp1a.1-/-* and *aqp8a. 1-/-* embryos at 24-26 hpf by analysing Golgi position relative to the nucleus. We were unable to analyze polarity in *aqp1a.1rk28/rk28*; *aqp8a.1rk29/rk29* embryos as they exist in an mCherry-containing transgenic zebrafish line (the Golgi marker is also tagged to mCherry). The results show that tip cell polarity is similar, if not more polarised, in *aqp1a.1-/-* and *aqp8a. 1-/-* embryos when compared to wildtype embryos (Fig. S10D). This new data is discussed in lines 234 to 237.

Figure 5:Panel D should be part of Figure 4.

Panel 5D is now in panel J of Figure 4 and described in lines 231 and 235.